# From Shortcut to Induction Head: How Data Diversity Shapes Algorithm Selection in Transformers

**Ryotaro Kawata**[*,1,2], **Yujin Song**[*,1,2], **Alberto Bietti**[3], **Naoki Nishikawa**[1,2],

**Taiji Suzuki**[1,2], **Samuel Vaiter**[4,5], **Denny Wu**[3,6]

[1]The University of Tokyo, [2]RIKEN AIP, [3]Flatiron Institute, [4]CNRS,
[5]Université Côte d'Azur, Laboratoire J.A. Dieudonné, [6]New York University

{kawata-ryotaro725,nishikawa-naoki259}@g.ecc.u-tokyo.ac.jp;
y.song.research@gmail.com; {abietti,dwu}@flatironinstitute.org;
taiji@mist.i.u-tokyo.ac.jp; samuel.vaiter@cnrs.fr

## Abstract

Transformers can implement both generalizable algorithms (e.g., induction heads) and simple positional shortcuts (e.g., memorizing fixed output positions). In this work, we study how the choice of pretraining data distribution steers a shallow transformer toward one behavior or the other. Focusing on a minimal trigger-output prediction task – copying the token immediately following a special trigger upon its second occurrence – we present a rigorous analysis of gradient-based training of a single-layer transformer. In both the infinite and finite sample regimes, we prove a transition in the learned mechanism: if input sequences exhibit sufficient diversity, measured by a low "max-sum" ratio of trigger-to-trigger distances, the trained model implements an induction head and generalizes to unseen contexts; by contrast, when this ratio is large, the model resorts to a positional shortcut and fails to generalize out-of-distribution (OOD). We also reveal a trade-off between the pretraining context length and OOD generalization, and derive the optimal pretraining distribution that minimizes computational cost per sample. Finally, we validate our theoretical predictions with controlled synthetic experiments, demonstrating that broadening context distributions robustly induces induction heads and enables OOD generalization. Our results shed light on the algorithmic biases of pretrained transformers and offer conceptual guidelines for data-driven control of their learned behaviors.

## 1 Introduction

Large language models (LLMs) leverage circuits of attention heads [VSP+17] to perform (implicit) algorithmic reasoning. Certain attention heads implement discrete algorithms — notably *induction heads* [ENO+21, OEN+22], which scan for previously seen token patterns in the context to predict subsequent tokens. Such heads enable *in-context learning* behaviors [BMR+20], allowing a transformer to continue a sequence such as $[A, B, \ldots, A] \rightarrow B$ purely by leveraging patterns in the context. By contrast, attention can also implement positional mechanisms that select tokens based solely on their location in the sequence [VTM+19, AWKA24]. These mechanisms can yield contrasting generalization performance [CBKZ24], and we expect the pretraining data distribution to play a central role in determining which mechanisms a model learns to rely on: depending on structural properties of the corpus, a transformer may either discover generalizable strategies (content-based retrieval) or adopt position-based shortcuts.

---

[*]Equal contribution.

39th Conference on Neural Information Processing Systems (NeurIPS 2025).

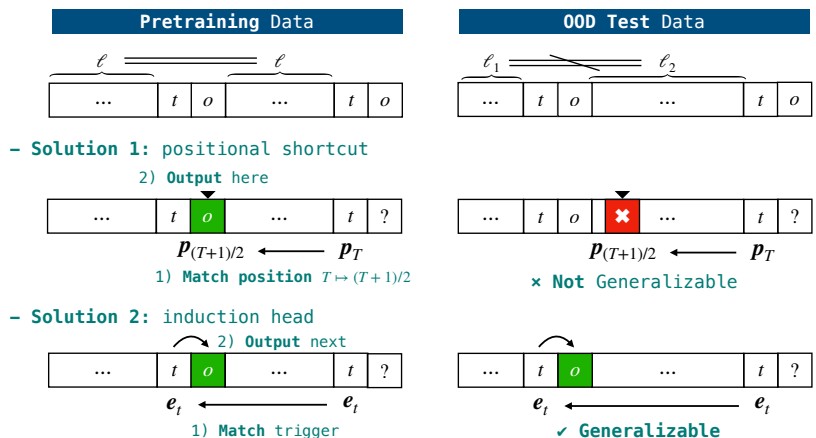

Figure 1: Two mechanisms for the associative copying task [..., t, o, ..., t] ↦ o. In the pretraining data, the size of irrelevant tokens before the occurrence of the first and second trigger $\ell$ remains fixed per sequence, hence allowing two solutions: $(i)$ *positional shortcut* that outputs the token at position $(T+1)/2$ for input length $T$; and $(ii)$ *induction head* using token embedding $e$, which finds the queried token and returns the ensuing token. Whereas on OOD sequences with varying $\ell_1 \neq \ell_2$, only $(ii)$ remains a valid solution.

**Motivation.** We theoretically study how pretraining data influences the implemented circuit and out-of-distribution (OOD) generalization performance of the transformer. This perspective is motivated from the empirical observation that pretrained models often leverage shortcut solutions that are brittle beyond the training distribution [MPL19, GJM+20, LAG+22]. For instance, a transformer might utilize the aforementioned position-based attention head to memorize that a certain output tends to occur at a particular position in the training text, instead of learning the underlying association (induction head); such positional shortcut is a double-edge sword in algorithmic tasks: transformers can achieve near-perfect accuracy in distribution, but struggle on test sequences of unseen lengths or structures. Since it is empirically known that the learned mechanism heavily depends on the structure of pretraining data [GTLV22, RLIGS22, RPCG23, WNB+25], we ask the following question.

*How does the data structure decide whether a pretrained transformer implements a generalizable mechanism (e.g.,induction head) or a shortcut that fails OOD (e.g., positional memorization)?*

## 1.1 Our Contributions

**Trigger-output Copying.** To investigate this question in a controlled setting, we introduce a minimal *trigger-output copying* task inspired by [BCB+23]. In this synthetic task, each input sequence contains a special trigger token that appears twice. The model must predict the token that immediately follows the *first* trigger when the trigger appears the second time. For example, given

$$\dots [\texttt{trigger}][X] \dots [\texttt{trigger}][?] \dots,$$

the correct prediction is $X$. Depending on the structure of the input sequence, this task admits multiple solutions. We focus on two mechanisms — see Figure 1.

- **Induction head.** The model attends back to the location of the previous trigger and copies the token following it; this works for arbitrarily long gaps between trigger occurrences (up to context-length limit).

- **Positional shortcut.** When the position of the first trigger is inferable from the second (e.g., under periodic structure), the model may copy the token using positional information alone. This shortcut is valid in-distribution but does not reflect the underlying association.

For this task, we define out-of-distribution (OOD) generalization as performance on test sequence with altered structure, where the trigger appears at positions not seen during pretraining (e.g., longer or aperiodic sequences). The induction head mechanism is robust to such shifts as it learns the correct association, whereas the positional shortcut typically fails OOD. Our goal is to identify a data-dependent transition between these two mechanisms that governs OOD generalization: intuitively, increasing the *diversity* of pretraining sequences — by varying the distances between trigger occurrences — dilutes positional signals and discourages the shortcut; conversely, as the number of

trigger tokens in the data grows, the effective signal for induction weakens. We make these intuitions precise in our theoretical analysis.

**Main Findings.** We provide a quantitative account of how pretraining data diversity shapes the mechanism learned by a pretrained transformer in the trigger-output copying task introduced above. Specifically, we rigorously analyze the in-distribution and out-of-distribution performance of a shallow (single-layer) transformer trained on this synthetic task. By studying an "early-phase" simplification of gradient descent in both the infinite-data (population loss) and finite-data (empirical loss) regimes, we show that the pretraining distribution directly selects the model's algorithm: when pretraining data are sufficiently "diverse" – as measured by a *max-sum ratio* of trigger distances – the transformer learns an induction head; when diversity is low, the model adopts a positional shortcut that fails to generalize OOD. Using this diversity measure that governs the phase transition, we discuss various tradeoffs and how to choose a pretraining distribution that induces the desired induction mechanism with minimal computational cost. Finally, we empirically probe the learned circuits by visualizing attention scores, and present evidence that a similar mechanism transition arises under standard gradient-based training beyond our theoretical setting.

## 1.2 Related Works

The induction head mechanism in transformers was first presented in the mechanistic interpretability literature [ENO+21, OEN+22], and followup theory investigates when such circuits emerge under simplified training dynamics and tasks [BCB+23, ETE+24, NDL24, Red24, CSWY24]. Empirical studies on algorithmic tasks (copying, arithmetic, sorting) demonstrate that transformers often rely on spurious "shortcut" solutions that fail to generalize, often due to poor use of positional information [ZBB+22, JdDE+23, ZBL+23, GJB+25]; the OOD brittleness of shortcut solution is also documented in [LAG+22]. A complementary thread links the structure of pretraining data to in-context behaviors: the function classes a transformer implements in context and the sensitivity of performance to data statistics such as corpus coverage and frequency [GTLV22, RLIGS22, MLH+22] or task diversity [RPCG23, LLZV+25]. Our analysis aligns with this view by making explicit how diversity in trigger distances steers the learned mechanism. Methodologically, we borrow the "early-phase" simplification of training dynamics and study the loss improvement after the first few gradient descent step [BES+22, DLS22, ORST23, BCB+23].

## 2 Problem Setting

**Notations.** For a positive integer $N$, we denote $[N] := \{1, 2, \ldots, N\}$. For integers $N_1 \leq N_2$, we define $[N_1 : N_2] := \{N_1, N_1 + 1, \ldots, N_2\}$. The Softmax function for an $N$-dimensional vector $\boldsymbol{v} \in \mathbb{R}^N$ is defined as $\text{Softmax}(\boldsymbol{v})_i := \frac{e^{v_i}}{\sum_{j=1}^{N} e^{v_j}}$. For a vector $\boldsymbol{v}$, we write $\boldsymbol{v} = O_2(f(N))$ if $\|\boldsymbol{v}\|_2 = O(f(N))$, and $\boldsymbol{v} = O_\infty(f(N))$ if $\max_i |v_i| = O(f(N))$. Similar notation is used for a matrix $\boldsymbol{A}$, where $\|\boldsymbol{A}\|_2$ and $\|\boldsymbol{A}\|_\infty$ denote its $\ell_2 \to \ell_2$ spectral and max norms, respectively.

### 2.1 Data Generating Process

We study the *trigger-output* setting to investigate how transformers acquire the induction head mechanism. Let $N \in \mathbb{N}$ denote the vocabulary size and $L \in \mathbb{N}$ the maximum input sequence length. We designate special tokens as *trigger tokens*. We define our data model as follows:

**Definition 1** (Data Distribution). *Let $\ell_1, \ell_2 \in \mathbb{N}$ such that $T := \ell_1 + \ell_2 + 3 \leq L - 1$. Let $N_{\text{trg}} \leq N$ denote the number of trigger tokens. A sequence $z_{1:T+1} \in [N]^{T+1}$ is sampled as follows:*

1. *Sample a trigger token $t \in [N_{\text{trg}}]$ and an output token $o \in [N_{\text{trg}} + 1 : N]$ uniformly at random, where $N_{\text{trg}} = o(N^{1/3})$.*

2. *Construct the sequence:*

$$z_{1:T+1} = (\ \underbrace{z_1, \ldots, z_{\ell_1}}_{\ell_1 \text{ irrelevant tokens}}, \quad \underbrace{t, o}_{\text{trigger-output pair}}, \quad \underbrace{z_{\ell_1+3}, \ldots, z_{\ell_1+\ell_2+2}}_{\ell_2 \text{ irrelevant tokens}}, \quad \underbrace{t, o}_{\text{trigger-output pair}}\ )$$

*where irrelevant token $z_i$ ($i \in [1 : \ell_1] \cup [\ell_1 + 3 : \ell_1 + \ell_2 + 2]$) is drawn i.i.d. from $[N_{\text{trg}} + 1 : N]$.*

*We refer to such a sequence as a* trigger-output *model with subtext lengths $\ell_1$ and $\ell_2$.*

In our data model, the task is to identify the output token $z_{T+1} = o$ from the sequence $z_{1:T} = (z_1, \ldots, z_{\ell_1}, t, o, z_{\ell_1+3}, \ldots, z_{\ell_1+\ell_2+2}, t)$. This can be achieved by implementing the *induction head mechanism* [ENO$^+$21, OEN$^+$22], which copies the token that follows the first occurrence of the trigger token and outputs it upon encountering the second occurrence of the same trigger.

Due to structure of the input sequence, transformer may also rely on positional shortcuts to achieve low loss; in particular, when the lengths of irrelevant tokens are identical within each sequence, i.e., $\ell_1 = \ell_2 = \ell$, a transformer can achieve 100% training accuracy simply by inferring the correct position to attend to $(T+1)/2 = \ell + 2$ from the position of the second trigger $T = 2\ell + 3$. Such positional solution does not make use of the semantic information and generally fails when $\ell_1 \neq \ell_2$.

To study the transition between the two mechanisms, we assume the pretraining data consists of a mixture of sequences with different lengths determined by $\ell = \ell_1 = \ell_2$.

**Definition 2.** *Consider a language model $p_{\boldsymbol{\theta}}(\cdot \mid z_1 z_2 \cdots z_T)$ that is pretrained on $M$ sequences $\left\{ z^{(i)}_{1:T^{(i)}+1} \right\}_{i=1}^M$ generated as follows:*

- *Sample $\ell^{(i)}$ from a distribution $\mathcal{D}_\ell$.*

- *Generate $z^{(i)}_{1:T^{(i)}+1}$ according to Definition 1 with $\ell_1 = \ell_2 = \ell^{(i)}$, i.e.,*

$$z^{(i)}_{1:T^{(i)}+1} = (z_1, \ldots, z_{\ell^{(i)}}, t, o, z_{\ell^{(i)}+3}, \ldots, z_{2\ell^{(i)}+2}, t, o).$$

**OOD Generalization.** Note that the pretraining distribution (defined by $\mathcal{D}_\ell$) may not cover all possible sequences. We say that $p_{\boldsymbol{\theta}}$ *generalizes out-of-distribution (OOD)* if it implements the correct copying mechanism across all possible $\ell$'s, that is, for any $\ell_1, \ell_2$ such that $\ell_1 + \ell_2 + 3 \leq L - 1$ (possibly $\ell_1 \neq \ell_2$), and for any test sequence $z_{1:T+1}$ generated from the trigger-output unigram model with subtext lengths $\ell_1$ and $\ell_2$ (Definition 1), we have

$$\arg \max_{k \in [N]} p_{\boldsymbol{\theta}}(k \mid z_1 z_2 \cdots z_T) = z_{T+1}.$$

## 2.2 Gradient-based Training of Single-layer Transformer

**Architecture and Embedding.** We consider a single-layer transformer block $f_{\mathrm{TF}}$ defined as

$$f_{\mathrm{TF}}(\boldsymbol{X}_{1:t}; \boldsymbol{W}_{KQ}, \boldsymbol{W}_V) = \boldsymbol{W}_V \boldsymbol{X}_{1:t} \operatorname{Softmax}(\boldsymbol{X}_{1:t}^\top \boldsymbol{W}_{KQ} \boldsymbol{x}_t) \in \mathbb{R}^N, \qquad (2.1)$$

where $\boldsymbol{W}_{KQ} \in \mathbb{R}^{D \times D}, \boldsymbol{W}_V \in \mathbb{R}^{N \times D}$ and $\boldsymbol{X}_{1:t} = (\boldsymbol{x}_1 \enspace \cdots \enspace \boldsymbol{x}_t) \in \mathbb{R}^{D \times t}$ denotes the input embeddings of $z_{1:t}$, with embedding dimension $D$. We define the embedding as follows:

**Definition 3.** *Let $D = L + 2N$. Let $\boldsymbol{p}_t \in \mathbb{R}^L$ denote the one-hot vector with a 1 at the $t$-th position (representing the positional embedding), and let $\boldsymbol{e}_z \in \mathbb{R}^N$ denote the one-hot vector with a 1 at the $z$-th position (representing the token identity).*

*We then construct the input embedding $\boldsymbol{x}_t$ as*

$$\boldsymbol{x}_t = \begin{bmatrix} \boldsymbol{p}_t \\ \boldsymbol{e}_{z_t} \\ \boldsymbol{e}_{z_{t-1}} \end{bmatrix} \in \mathbb{R}^{L+2N}. \qquad (2.2)$$

The prediction probability is given by

$$p_{(\boldsymbol{W}_{KQ}, \boldsymbol{W}_V)}(z_{T+1} = k \mid z_1 \cdots z_T) = [\operatorname{Softmax}(f_{\mathrm{TF}}(\boldsymbol{X}_{1:t}; \boldsymbol{W}_{KQ}, \boldsymbol{W}_V))]_k.$$

**Remark 1.** *We make the following remarks on the design of our architecture and embedding.*

- *The architecture (with the FFN is absorbed into the value matrix $\boldsymbol{W}_V$, and tied key and query projections) is commonly used in theoretical analyses and mechanistic studies [LLR23, BCB$^+$23, NDL24]; the simplification allows us to focus on the inductive bias by simple attention mechanisms, while retaining sufficient expressiveness to implement algorithmic behaviors.*

- *Two-layer architecture is typically needed to implement the induction head mechanism, where the first layer often learns to detect the trigger and identify of the following token via attention to the previous token [SHT24]. To reflect this inductive step in our simplified single-layer setting, we explicitly encode the identity of the previous token $z_{t-1}$ in the third component of the embedding $\boldsymbol{x}_t$. This choice also echoes recent empirical developments that incorporate information of previous tokens directly into the current state, such as Mamba [GD23], RWKV [PAA$^+$23], and convolution augmentations [LZHO25, All25].*

---

**Algorithm 1:** Gradient-based training of single-layer transformer

---

**Input** : Learning rate $\eta_{KQ}, \eta_V$
**Initialize** $\boldsymbol{W}_{KQ}(0) = \boldsymbol{O}_{(L+2N)\times(L+2N)}, \boldsymbol{W}_V(0) = \boldsymbol{O}_{N\times(L+2N)}$
**Gradient descent on** $\boldsymbol{W}_V$
$\qquad \boldsymbol{W}_V(1) \leftarrow \boldsymbol{W}_V(0) - \eta_V \nabla_{\boldsymbol{W}_V} \frac{1}{M_V} \sum_{i=1}^{M_V} \mathcal{L}(\boldsymbol{X}_{1:T^{(i)}}^{(i)}; \boldsymbol{W}_{KQ}(0), \boldsymbol{W}_V(0))$
**Gradient descent on** $\boldsymbol{W}_{KQ}$
$\qquad \boldsymbol{W}_{KQ}(1) \leftarrow \boldsymbol{W}_{KQ}(0) - \eta_{KQ} \nabla_{\boldsymbol{W}_{KQ}} \frac{1}{M_{KQ}} \sum_{i=M_V+1}^{M_V+M_{KQ}} \mathcal{L}(\boldsymbol{X}_{1:T^{(i)}}^{(i)}; \boldsymbol{W}_{KQ}(0), \boldsymbol{W}_V(1))$
**Output:** Prediction $f_{\mathrm{TF}}(\cdot)$

---

**Gradient-based Learning Algorithm.** We use gradient descent (Algorithm 1) on the cross-entropy loss to pretrain our shallow transformer (2.1),

$$\mathcal{L}(\boldsymbol{X}_{1:T^{(i)}}^{(i)}; \boldsymbol{W}_{KQ}, \boldsymbol{W}_V) = \mathrm{CrossEntropy}(\boldsymbol{e}_{z_{T^{(i)}+1}}, \mathrm{Softmax}(f_{\mathrm{TF}}(\boldsymbol{X}_{1:T^{(i)}}; \boldsymbol{W}_{KQ}, \boldsymbol{W}_V))).$$

In Algorithm 1, we apply a *single gradient descent step* with large learning rate on the value and key-query matrices. This is motivated by recent studies [ORST23, BCB+23, WS24] showing that the first gradient step can induce associative memory tied to specific components of the input embedding. In particular, the gradient can often be expressed as a linear combination of outer products $\boldsymbol{w}\boldsymbol{v}^\top$, where either $\boldsymbol{w}$ or $\boldsymbol{v}$ corresponds to embedding vectors such as $\boldsymbol{e}_{z_t}, \boldsymbol{e}_{z_{t-1}}$, or $\boldsymbol{p}_t$. Such a gradient structure is sufficient to construct simple forms of associative memory within the model. We remark that similar single-step update is commonly used in the analysis of feature learning in shallow neural networks [BES+22, DLS22, BEG+22] and transformers [OSSW24, NSO+25, WNB+25].

## 3 Main Result: Data-driven Transition Between Mechanisms

### 3.1 Positional Shortcut vs. Induction Head

In this section, we illustrate how the diversity of pretraining distribution influences which algorithm the trained transformer implements — either the positional shortcut or the induction head. The following quantity plays a central role in our characterization.

**Definition 4.** *For each $\ell$, let $q_\ell$ denote the probability mass assigned under $\mathcal{D}_\ell$, and $\mathcal{S}$ the support of $\mathcal{D}_\ell$. We define the max-sum ratio as*

$$R(\mathcal{D}_\ell) = \frac{\max_{\ell\in\mathcal{S}} \ell^{-1}q_\ell}{\sum_{\ell\in\mathcal{S}} \ell^{-1}q_\ell}.$$

**Interpretation of max-sum ratio.** The max-sum ratio can be seen as a *diversity* measure of $\mathcal{D}_\ell$. The following example provides an intuitive illustration:

**Example 1.** *Let $\mathcal{D}_\ell = \mathrm{Unif}(\{\ell_0, \ell_0 + 1, \ldots, \ell_0 + K - 1\})$. Then the max-sum ratio is given by*

$$R(\ell_0, K) = \frac{\ell_0^{-1}}{\sum_{k=0}^{K-1}(\ell_0 + k)^{-1}}, \tag{3.1}$$

*which monotonically decreases with $K$; hence greater diversity of $\mathcal{D}_\ell$ gives smaller max-sum ratio.*

Note that the max-sum ratio does not merely capture the width of the distribution: in Example 1, increasing $\ell_0$ while keeping $K$ fixed decreases the proportion of $\ell_0^{-1}$ in $[\ell_0^{-1}, \ldots, (\ell_0 + K - 1)^{-1}]$, thus reducing the max-sum ratio. Hence, even with a narrow range, shifting the distribution rightward – placing more probability on larger $\ell$ – naturally yields a smaller max-sum ratio. This is because the max-sum ratio weights each probability mass $q_\ell$ by $\ell^{-1}$.

**Learning under Population Loss.** The next theorem shows the existence of a threshold in the max-sum ratio that determines whether OOD generalization is achieved, in the infinite-data limit.

**Theorem 5** (Infinite Sample Setting). *Suppose we run Algorithm 1 on the expected loss $\mathbb{E}[\mathcal{L}(\boldsymbol{X}_{1:T}; \boldsymbol{W}_{KQ}, \boldsymbol{W}_V)]$ with learning rates $\eta_V \lesssim 1, \eta_V\eta_{KQ} \gtrsim \frac{N^3}{N_{\mathrm{trg}}^3}\log N$. Then, there exist $\epsilon_1(N_{\mathrm{trg}}), \epsilon_2(N_{\mathrm{trg}}) = \Theta(N_{\mathrm{trg}}^{-1})$ such that:*

- *If $R(\mathcal{D}_\ell) < \epsilon_1$, then the pretrained transformer generalizes OOD, as defined in Definition 2.*

- *If $R(\mathcal{D}_\ell) > \epsilon_2$, then there exist OOD test sequences such that the pretrained transformer fails.*

**Remark 2.**

- *Note that the training data only contain sequences with $\ell_1 = \ell_2$, and thus a positional shortcut (as illustrated in Figure 1) can still achieve 100% training accuracy. However, since the OOD test data include sequences with $\ell_1 \neq \ell_2$, such shortcuts inevitably fail. Our main theorems show that the pretrained transformer avoids such shortcuts when the max-sum ratio is below a certain threshold, i.e., when the data distribution is sufficiently diverse.*

- *We also provide a tight $\Theta(N_{\mathrm{trg}}^{-1})$ characterization of the max-sum ratio threshold, indicating that increasing the number of possible triggers makes OOD generalization more difficult. The underlying mechanism is discussed in the ensuing subsection.*

**Learning under Empirical Loss.**  Our next result establishes (via gradient concentration) similar transition behavior in the finite-sample setting.

**Theorem 6** (Finite Sample Setting). *Suppose we run Algorithm 1 with the same learning rate scaling as in Theorem 5, and with sample sizes $M_{KQ} \gtrsim \text{poly} \log N \cdot \frac{N^3}{N_{\mathrm{trg}}^2}\left(\sum_\ell \sqrt{q_\ell}\right)^2$ and $M_V \gtrsim \text{poly} \log N \cdot \frac{N^5}{N_{\mathrm{trg}}^2}\left(\frac{\sum_{\ell \in \mathcal{S}} \sqrt{q_\ell}}{\sum_{\ell \in \mathcal{S}} q_\ell \ell^{-1}}\right)^2$. Then, with probability at least 0.99 there exist $\epsilon_1'(N_{\mathrm{trg}}), \epsilon_2'(N_{\mathrm{trg}}) = \Theta(N_{\mathrm{trg}}^{-1})$ such that the assertion of Theorem 5 holds by substituting $(\epsilon_1', \epsilon_2')$ for $(\epsilon_1, \epsilon_2)$.*

## 3.2  Mechanism of Algorithm Selection

Now we take a closer look at how the *positional shortcut* and the *induction head* are implemented in the attention. We begin with the case where the support of $\mathcal{D}_\ell$ is a singleton and $N_{\mathrm{trg}} = 1$. After a single gradient step, the parameter matrix $\boldsymbol{W}_{KQ}$ can be shown to implement a form of associative memory over the relevant embedding vectors.

**Lemma 7** (Informal). *Let $\mathcal{D}_\ell = \{\ell\}$, and assume the trigger consists of a single token $w$. After one gradient step of Algorithm 1, $\boldsymbol{W}_{KQ}$ takes the form*

$$\boldsymbol{W}_{KQ} \propto T(\ell)^{-1} \begin{bmatrix} (\boldsymbol{p}_{\ell+2} + \boldsymbol{p}_{\ell+3}) \\ \boldsymbol{0} \\ \boldsymbol{e}_w \end{bmatrix} \begin{bmatrix} \boldsymbol{p}_{T(\ell)}^\top & \boldsymbol{e}_w^\top & \boldsymbol{0} \end{bmatrix},$$

*where $T(\ell) = 2\ell + 3$ denotes the position of the second occurrence of the trigger token.*

To further simplify the exposition, we ignore the cross terms between $\boldsymbol{p}$ and $\boldsymbol{e}$ and assume that $\boldsymbol{W}_{KQ}$ takes the following form:

$$\boldsymbol{W}_{KQ} \propto \underbrace{T(\ell)^{-1} \begin{bmatrix} (\boldsymbol{p}_{\ell+2} + \boldsymbol{p}_{\ell+3}) \\ \boldsymbol{0} \\ \boldsymbol{0} \end{bmatrix} \begin{bmatrix} \boldsymbol{p}_{T(\ell)}^\top & \boldsymbol{0}^\top & \boldsymbol{0}^\top \end{bmatrix}}_{\text{positional shortcut}} + \underbrace{T(\ell)^{-1} \begin{bmatrix} \boldsymbol{0} \\ \boldsymbol{0} \\ \boldsymbol{e}_w \end{bmatrix} \begin{bmatrix} \boldsymbol{0} & \boldsymbol{e}_w^\top & \boldsymbol{0} \end{bmatrix}}_{\text{induction head}} \quad (3.2)$$

Now consider an OOD test sequence as in Figure 1, whose total length matches the training sequence but whose first and second subtext lengths differ: $\ell_1 + \ell_2 = 2\ell, \ell_1 \neq \ell_2$. In this case, the two terms in (3.2) contribute to the attention score

$$\text{Softmax}\left(\boldsymbol{X}_{1:T_{\mathrm{test}}}^\top \boldsymbol{W}_{KQ} \boldsymbol{x}_{T_{\mathrm{test}}}\right) \quad \text{with} \quad T_{\mathrm{test}} = \ell_1 + \ell_2 + 3 = T(\ell)$$

as follows (see Figure 2), noting that $\boldsymbol{x}_{T_{\mathrm{test}}} = \begin{bmatrix} \boldsymbol{p}_{T(\ell)} & \boldsymbol{e}_w & * \end{bmatrix}^\top$:

- **1st term (positional shortcut).** Regardless of $\ell_1$, it attends to the positions $\ell + 2 = (T_{\mathrm{test}} + 1)/2$ and $\ell + 3 = (T_{\mathrm{test}} + 3)/2$. In particular, for the former, even though $\ell_1 \neq \ell_2$, the transformer incorrectly associates the second trigger position $T_{\mathrm{test}}$ with $(T_{\mathrm{test}} + 1)/2$ as if $\ell_1 = \ell_2$.[2]

---

[2] For the latter position $(T_{\mathrm{test}} + 3)/2$, the model also attends to the same token via the previous-token embedding. This follows from a detailed computation of $\boldsymbol{W}^V$, which we omit here.

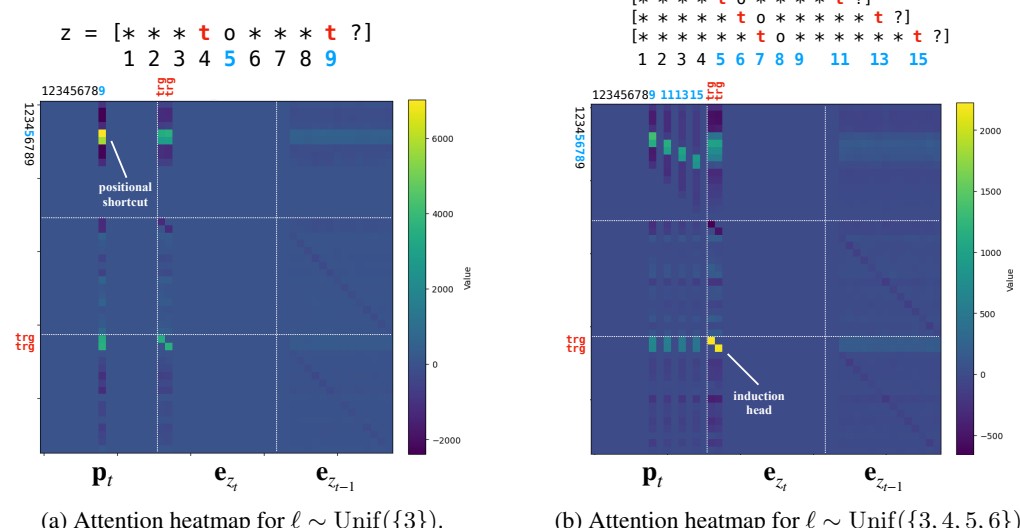

(a) Attention heatmap for $\ell \sim \mathrm{Unif}(\{3\})$.      (b) Attention heatmap for $\ell \sim \mathrm{Unif}(\{3, 4, 5, 6\})$.

Figure 2: Attention heatmaps of $\boldsymbol{W}^{KQ}$ when the pretraining sequence diversity is small (left) and large (right). In the left figure, there is a strong *positional shortcut* that links position 9 to position 5 (the correct position in pretraining data), whereas in the right figure, the trigger positions are more dispersed, weakening this shortcut. Instead, a signal corresponding to *induction head* – detecting tokens after trigger – becomes dominant.

- **2nd term (induction head).** It attends to tokens whose third embedding block equals $\boldsymbol{e}_w$, i.e., tokens whose *previous* token is the trigger $w$. In other words, it scans for the trigger $w = z_{T_{\mathrm{test}}}$ and then attends to its *next* token — this is precisely the desired induction head behavior.

Thus, the learned attention matrix implements a mixture of positional shortcut and induction head, and the relative strength of these components determines which algorithm is ultimately selected. Two factors affect this balance: the diversity of irrelevant token length $\ell$ and the trigger size $N_{\mathrm{trg}}$.

**Length distribution $\mathcal{D}_\ell$.** Equation (3.2) describes the case where $\ell$ is deterministic. When $\ell$ is distributed according to $\mathcal{D}_\ell$, $\boldsymbol{W}_{KQ}$ becomes a superposition over $\ell$:

$$\boldsymbol{W}_{KQ}(1) \propto \sum_\ell q_\ell \, T(\ell)^{-1} \begin{bmatrix} (\boldsymbol{p}_{\ell+2} + \boldsymbol{p}_{\ell+3}) \\ \mathbf{0} \\ \mathbf{0} \end{bmatrix} \begin{bmatrix} \boldsymbol{p}_{T(\ell)}^\top & \mathbf{0}^\top & \mathbf{0}^\top \end{bmatrix} + \mathbb{E}[T(\ell)^{-1}] \begin{bmatrix} \mathbf{0} \\ \mathbf{0} \\ \boldsymbol{e}_w \end{bmatrix} \begin{bmatrix} \mathbf{0} & \boldsymbol{e}_w^\top & \mathbf{0} \end{bmatrix}.$$

Here, the first term spreads its mass across multiple positions and is consequently weakened, whereas the second term does not depend on $\ell$ and retains its strength. As a result, the magnitude of the former is at most $\max_\ell q_\ell \, T(\ell)^{-1}$, while that of the latter is $\sum_\ell q_\ell \, T(\ell)^{-1}$. Since $T(\ell) \asymp \ell$, the ratio between the strengths of positional memory and the induction head is nothing but the max-sum ratio $R(\mathcal{D}_\ell)$. This explains why the max-sum ratio governs algorithm selection.

**Trigger size $N_{\mathrm{trg}}$.** In (3.2), when the trigger size $N_{\mathrm{trg}} \geq 2$, the second term is replaced by

$$N_{\mathrm{trg}}^{-1} \sum_{w \in [N_{\mathrm{trg}}]} T(\ell)^{-1} \begin{bmatrix} \mathbf{0} \\ \mathbf{0} \\ \boldsymbol{e}_w \end{bmatrix} \begin{bmatrix} \mathbf{0} & \boldsymbol{e}_w^\top & \mathbf{0} \end{bmatrix},$$

while the first term remains unchanged. Hence, the induction-head signal is split across trigger types and its strength decreases proportionally to $N_{\mathrm{trg}}^{-1}$. This explains $\Theta(N_{\mathrm{trg}}^{-1})$ threshold in Theorem 5.

The above intuition is visualized in an experiment reported in Figure 2.

**Example 2.** *In Figure 2, we set $N = 16$ and $N_{\mathrm{trg}} = 2$, train the model with $\mathcal{D}(\ell) = 3$ and $\mathcal{D}(\ell) = \mathrm{Unif}([3:8])$, and visualize the resulting $\boldsymbol{W}^{KQ}$. The trigger-token set is $\{1, 2\}$. The training setting is the same as that in Section 4.1.*

- **(Left):** *when $\mathcal{D}(\ell) = \{3\}$, $\boldsymbol{W}^{KQ}$ has a strong component that maps position 9 to position 5. Although it also contains an induction head component that maps between trigger tokens, it is comparatively weak compared to the positional signal.*

- **(Right):** *when $\mathcal{D}(\ell) = \mathrm{Unif}([3:8])$, $\boldsymbol{W}^{KQ}$ exhibits a superposition of signals mapping position $k$ to $(k+1)/2$, which results in each individual signal being weakened. In contrast, the induction head signal does not diminish.*

### 3.3 Tradeoff between Context Length and OOD Generalization

As discussed in Section 3.1, the max-sum ratio captures not only the overall "width" of the distribution but also decreases as mass shifts toward larger $\ell$. This effect becomes especially pronounced near the $\Theta(N_{\mathrm{trg}}^{-1})$ threshold identified in Theorems 5 and 6:

**Example 3.** *Consider the max-sum ratio for the uniform distribution (3.1). If $\ell_0 = 1$, then $R(\ell_0, K) = \Theta((\log K)^{-1})$. To attain a max-sum ratio of order $O(N_{\mathrm{trg}}^{-1})$ – the OOD generalization threshold in Theorems 5 and 6 – the support width must satisfy $K \gtrsim \exp(N_{\mathrm{trg}})$. By contrast, if $\ell_0 = \Theta(N_{\mathrm{trg}})$, then it suffices to take $K = \Theta(N_{\mathrm{trg}})$ to obtain a max-sum ratio of $O(N_{\mathrm{trg}}^{-1})$.*

Therefore, merely "widening" the distribution may not be efficient to reduce the max-sum ratio; biasing pretraining toward longer contexts is substantially more effective. This, in turn, suggests that reliably learning the induction-head mechanism (and hence achieving OOD generalization) may incur greater computational cost due to longer training sequences.

We now consider the "optimal" shape of the pretraining sequence (under the constraint in Definition 2) that learns the induction-head mechanism with minimal compute. Since the forward-pass cost scales quadratically with context length, we seek short contexts while maintaining a favorable max-sum ratio. Formally, for $U \geq N_{\mathrm{trg}}$, consider the optimization problem

$$\mathbb{P}: \begin{cases} \text{minimize} & \sum_{\ell=1}^{U} q_\ell \ell^2 \\ \text{subject to} & \frac{\max_{\ell=1}^{U} q_\ell \ell^{-1}}{\sum_{\ell=1}^{U} q_\ell \ell^{-1}} \leq N_{\mathrm{trg}}^{-1} \\ & \sum_{\ell=1}^{U} q_\ell = 1 \\ & q_1, \ldots, q_U \geq 0 \end{cases}$$

This objective is the sample-average forward-pass cost in pretraining; the constraints enforce the OOD threshold from Theorem 7 and the normalization of $(q_\ell)_{\ell=1}^{U}$. This problem is a linear program whose optimizer is characterized below.

**Proposition 8.** *The optimal solution of problem $\mathbb{P}$ assigns linearly increasing probability mass to the first $N_{\mathrm{trg}}$ context lengths and zero to the remaining ones:*

$$(q_1, q_2, \ldots, q_U) = Z^{-1}(1, 2, \ldots, N_{\mathrm{trg}}, 0, \ldots, 0),$$

*where the normalization constant is $Z = N_{\mathrm{trg}}(N_{\mathrm{trg}} + 1)/2$.*

In other words, to minimize average forward-pass cost per sample while meeting the OOD generalization constraint, the pretraining distribution should be linear in the context length, making $q_\ell \ell^{-1}$ uniform over $\ell \leq N_{\mathrm{trg}}$. We note that if one optimizes a different objective (e.g., incorporating sample complexity), the optimal pretraining distribution may change.

## 4 Numerical Experiments

### 4.1 Experiments for Theoretical Setting

To observe the transition from positional shortcut to induction head, we first consider the architecture defined in (2.1) and conduct experiments under the data model described in Definition 1.

#### 4.1.1 Experimental Setup

**Dataset.** We generate training and test data according to the trigger-output setting in Definition 1.

- In the *pretraining data*, the lengths of irrelevant tokens $\ell_1$, $\ell_2$ are always equal. We choose two integers $\ell_{\min}$ and $\ell_{\max}$ ($\ell_{\min} \leq \ell_{\max}$), and length $\ell$ is sampled from $\mathrm{Unif}([\ell_{\min}, \ell_{\max}])$.

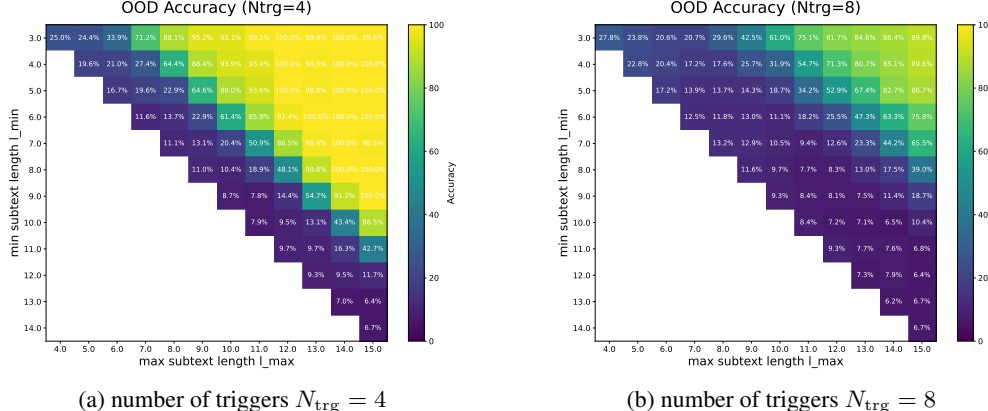

(a) number of triggers $N_{\text{trg}} = 4$        (b) number of triggers $N_{\text{trg}} = 8$

Figure 3: Out-of-distribution accuracy map over varying $\ell_{\min}$ (vertical) and $\ell_{\max}$ (horizontal); moving right indicates greater diversity of $\ell$ in the pretraining distribution.

- In the *OOD test data*, we shift the position of the first trigger to produce non-periodic sequences. Specifically, we first sample $\ell \sim \text{Unif}([\ell_{\min} + 1, \ell_{\max}])$, and then sample $\ell_1 \sim \text{Unif}(\{1, \ldots, 2\ell - 1\} \setminus \{\ell\})$, defining $\ell_2 = 2\ell - \ell_1$ so that $\ell_1 \neq \ell_2$.

**Model architecture, embedding, and training.** We implement a one-layer transformer architecture as defined in (2.1) with embeddings defined in (2.2). Training follows Algorithm 1, and the learning rates for $\boldsymbol{W}^V$ and $\boldsymbol{W}^{KQ}$ are set to $10^3$ and $10^4$, respectively. Both matrices are trained with the empirical cross entropy loss computed on 8192 training examples.

#### 4.1.2 Empirical Observations

**OOD Accuracy.** We conduct experiments for all combinations of $\ell_{\min} \in [3, 15]$ and $\ell_{\max} \in [3, 15]$ such that $\ell_{\min} < \ell_{\max}$, and evaluate all models on 1024 OOD test samples. The test accuracies (with different trigger size $N_{\text{trg}}$) are presented in Figure 3.

- OOD accuracy tends to increase as $\ell_{\max}$ increases (with $\ell_{\min}$ fixed). This suggests that a greater diversity in the training data biases the model towards the induction head.

- Comparing the left and right figures, we see that as the trigger size increases, the region where OOD generalization is achieved shifts rightward, suggesting an increased difficulty of induction head learning with larger $N_{\text{trg}}$, as predicted by Theorem 6.

**Error Visualization.** Our theory predicts two characteristic error modes:

- *Pseudo trigger position.* For non-periodic OOD evaluation data with $\ell_1 + \ell_2 = 2\ell$ and $\ell_1 \neq \ell_2$, let $\tilde{\ell} = (\ell_1 + \ell_2)/2$. The positional shortcut maps the second-trigger position $\ell_1 + \ell_2 + 3$ to the *pseudo* output position $\tilde{\ell} + 2$. Accordingly, we measure the fraction of instances where the model outputs $z_{\tilde{\ell}+2}$ and report this frequency as the *pseudo accuracy rate*.

- *Leftmost position.* Since the leftmost trigger in the pretraining data typically provides the strongest positional signal, the model may output $z_{\ell_{\min}+2}$ *independent of the second trigger position*. This error mode is especially likely when $N_{\text{trg}}$ is small. We record its frequency as the *leftmost rate*.

Figure 5 in Appendix E illustrates the existence of these positional shortcuts. We observe that the error rate due to the pseudo-trigger mechanism is higher near the diagonal, and both errors decline as $\ell_{\max}$ increases.

### 4.2 Experiments for Practical Settings

Next we examine whether a similar transition from positional shortcut to induction head occurs in more standard gradient-based pretraining beyond our theoretical simplification. We consider a three-layer transformer architecture with separated key-query matrices, MLPs, and residual connections, where all parameters are learned *jointly* using the AdamW optimizer [KB14, LH19]. The dataset

is generated in the same way as in Section 4.1: we set $N = 32$ and $N_{\mathrm{trg}} = 1$, and varied $\ell_{\min} = 4, 8, \ldots, 20$, $\ell_{\max} = 4, 8, \ldots, 40$. More experimental details can be found in Appendix E.

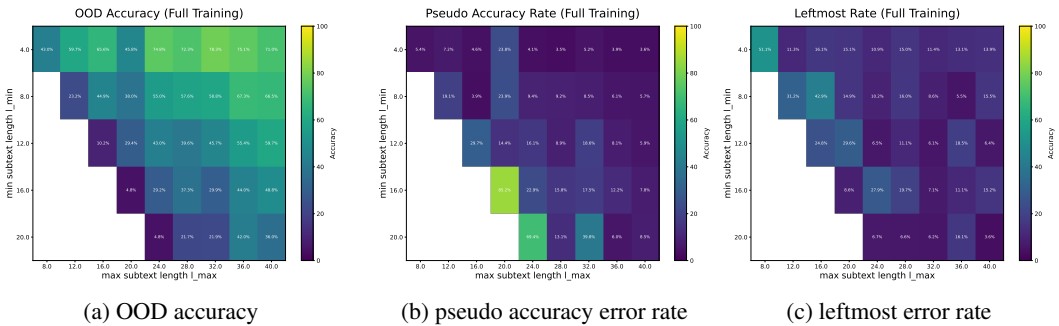

(a) OOD accuracy         (b) pseudo accuracy error rate         (c) leftmost error rate

Figure 4: Accuracy map over $\ell_{\min}$ (vertical), $\ell_{\max}$ (horizontal) for a 3-layer transformer trained with AdamW.

**Empirical Observations.** Figure 4 shows the OOD accuracy, pseudo accuracy rate, and leftmost rate, following the same setup as in Section 4.1. Note that as the diversity of the pretraining distribution increases, the OOD generalization accuracy improves and the errors due to the positional shortcut decrease — this is consistent with our theoretical prediction in Section 3. We also observe that the transition point is less sharp compared to our theoretical setting.

## 5 Conclusion

In this work, using a simplified trigger–output task, we developed a theoretical analysis showing that gradient-based training implicitly selects between two distinct mechanisms with different out-of-distribution generalization properties — an induction head or a positional shortcut. We introduced the *max-sum ratio* as a key quantity governing this selection. Our results demonstrate that the statistical structure of pretraining data critically shapes the algorithms internalized by transformers, offering quantitative insights into steering learning via data design.

We conclude with several directions for future work. First, beyond absolute positional embeddings, it is important to characterize which positional shortcuts can arise under relative position embeddings and related variants. Second, while our analysis centers on a single-layer architecture, a two-layer model naturally delegates retrieval to the first layer (recovering the token corresponding to $e_{z_{t-1}}$); analyzing the coupled dynamics that emerge from this decomposition is an intriguing next step. Finally, developing methods to analyze and quantify richer classes of algorithmic biases – beyond the induction–shortcut dichotomy – would deepen our understanding of how pretraining distributions induce specific computational circuits.

### Acknowledgements

RK was partially supported by JST CREST (JPMJCR2115). NN was partially supported by JST ACT-X (JPMJAX24CK) and JST BOOST (JPMJBS2418). TS was partially supported by JSPS KAKENHI (24K02905) and JST CREST (JPMJCR2015). This research is supported by the National Research Foundation, Singapore, Infocomm Media Development Authority under its Trust Tech Funding Initiative, and the Ministry of Digital Development and Information under the AI Visiting Professorship Programme (award number AIVP-2024-004). Any opinions, findings and conclusions or recommendations expressed in this material are those of the author(s) and do not reflect the views of National Research Foundation, Singapore, Infocomm Media Development Authority, and the Ministry of Digital Development and Information.

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

## A  Preliminaries

Throughout the paper, $O(\cdot)$ notation is taken with respect to the vocabulary size $N$ and we assume the following scaling:

**Assumption 1** (Scaling between problem parameters). *Number of triggers $N_{\mathrm{trg}}$ satisfies $N_{\mathrm{trg}} = o(N^{1/3})$. Context length $T$ satisfies $T = o(N/N_{\mathrm{trg}}^2)$ almost surely.*

We also assume the following:

**Assumption 2.** *We assume $\ell \geq 4$ almost surely at pretraining and $L = \mathrm{poly}(N)$.*

$\mathbb{I}(A)$ denotes the indicator function of the event $A$, that is, $\mathbb{I}(A) = 1$ if $A$ holds, and $0$ otherwise. For $a \leq b$, $\mathbf{1}_{a:b}$ denotes the vector whose $a, a+1, \ldots, b$-th entries are one and others are zero. For a matrix $\boldsymbol{A} \in \mathbb{R}^{m \times n}$, we use the notation $\boldsymbol{A}[:, I:J]$ to denote the submatrix consisting of all rows and columns from index $I$ to $J$. For an arbitrary matrix $\boldsymbol{A}$, $\lambda_i(\boldsymbol{A})$ is the $i$-th largest eigenvalue of $\boldsymbol{A}$.

## B  Infinite Sample Analysis

In this section, we analyze Algorithm 1 with infinite sample size. Recall that we defined

$$f_{\mathrm{TF}}(\boldsymbol{X}_{1:t}; \boldsymbol{W}_{KQ}, \boldsymbol{W}_V) = \boldsymbol{W}_V \boldsymbol{X}_{1:t} \mathrm{Softmax}(\boldsymbol{X}_{1:t}^\top \boldsymbol{W}_{KQ} \boldsymbol{x}_t) \in \mathbb{R}^N$$

and next token prediction loss

$$\mathcal{L}(\boldsymbol{X}_{1:T^{(i)}}^{(i)}; \boldsymbol{W}_{KQ}, \boldsymbol{W}_V) = \mathrm{CrossEntropy}(\boldsymbol{e}_{z_{T^{(i)}+1}}, \mathrm{Softmax}(f_{\mathrm{TF}}(\boldsymbol{X}_{1:T^{(i)}}; \boldsymbol{W}_{KQ}, \boldsymbol{W}_V))).$$

For simplicity, we denote the population loss as $\bar{\mathcal{L}}(\boldsymbol{W}_{KQ}, \boldsymbol{W}_V) := \mathbb{E}_T[\mathbb{E}_{\boldsymbol{X}_{1:T}}[\mathcal{L}(\boldsymbol{X}_{1:T}; \boldsymbol{W}_{KQ}, \boldsymbol{W}_V)]]$.

### B.1  Population Gradient of $\mathbf{W}_V$

Note that $\mathrm{Softmax}(\boldsymbol{X}_{1:T}^\top \boldsymbol{W}_{KQ} \boldsymbol{x}_T) = [1/T, \ldots, 1/T]^\top$ and $\mathrm{Softmax}(f_{\mathrm{TF}}(\boldsymbol{X}_{1:t}; \boldsymbol{W}_{KQ}, \boldsymbol{W}_V)) = [1/N, \ldots, 1/N]^\top$ is satisfied at initialization, for any $\boldsymbol{X}_{1:t}$. From [BCB$^+$23, Lemma 1], the population loss can be calculated as

$$
\begin{aligned}
&\nabla_{\boldsymbol{W}_V} \bar{\mathcal{L}}(\boldsymbol{W}_{KQ}, \boldsymbol{W}_V) \\
&= \frac{1}{N} \sum_{k=1}^N \boldsymbol{e}_k \mathbb{E}_T\left[\frac{1}{T} \sum_{t=1}^T \mathbb{E}[\boldsymbol{x}_t]^\top\right] - \sum_{k=N_{\mathrm{trg}}+1}^N \boldsymbol{e}_k \mathbb{E}_T\left[\frac{1}{T} \sum_{t=1}^T \mathbb{E}[\mathbb{I}(z_{T+1} = k)\boldsymbol{x}_t^\top]\right]. \\
&= \frac{1}{N} \sum_{k=1}^N \boldsymbol{e}_k \mathbb{E}_T\left[\frac{1}{T} \sum_{t=1}^T \mathbb{E}[\boldsymbol{x}_t]^\top\right] - \frac{1}{N - N_{\mathrm{trg}}} \sum_{k=N_{\mathrm{trg}}+1}^N \boldsymbol{e}_k \mathbb{E}_T\left[\frac{1}{T} \sum_{t=1}^T \mathbb{E}[\boldsymbol{x}_t | z_{T+1} = k]^\top\right].
\end{aligned}
$$

We conduct block-wise calculation for the population gradient: let

$$\boldsymbol{W}_V = \left[\boldsymbol{W}_V^{(1)}, \boldsymbol{W}_V^{(2)}, \boldsymbol{W}_V^{(3)}\right]$$

and let

$$\boldsymbol{W}_V^* = \left[\boldsymbol{W}_V^{*,(1)}, \boldsymbol{W}_V^{*,(2)}, \boldsymbol{W}_V^{*,(3)}\right]$$

be $\boldsymbol{W}_V$ after one GD step, i.e., $\boldsymbol{W}_V^* = -\eta_V \nabla_{\boldsymbol{W}_V} \bar{\mathcal{L}}(\boldsymbol{W}_{KQ}, \boldsymbol{W}_V)$, where $\boldsymbol{W}_V^{(1)} \in \mathbb{R}^{N \times D}$, $\boldsymbol{W}_V^{(2)}, \boldsymbol{W}_V^{(3)} \in \mathbb{R}^{N \times N}$. In this section we show the following, using the rescaling $\eta_V = N\tilde{\eta}_V$ for notation simplicity.

**Lemma 9.** *If we use stepsize $N\tilde{\eta}_V$ for $\boldsymbol{W}_V$, then it holds that*

$$\langle \boldsymbol{e}_k, \boldsymbol{W}_V^{*,(1)} \boldsymbol{p}_t \rangle = \begin{cases} -\alpha_t \tilde{\eta}_V & (k \in [N_{\mathrm{trg}}]), \\ \frac{\alpha_t \tilde{\eta}_V N_{\mathrm{trg}}}{N - N_{\mathrm{trg}}} & (k \notin [N_{\mathrm{trg}}]), \end{cases} \tag{B.1}$$

$$\langle e_j, W_V^{*,(2)} e_k \rangle = \begin{cases} -2\tilde{\eta}_V \mathbb{E}[T^{-1}] N_{\mathrm{trg}}^{-1} & (j, k \in [N_{\mathrm{trg}}]) \\ -\tilde{\eta}_V \frac{1 - 2\mathbb{E}[T^{-1}]}{N - N_{\mathrm{trg}}} & (j \in [N_{\mathrm{trg}}], k \notin [N_{\mathrm{trg}}]) \\ 2\tilde{\eta}_V \frac{\mathbb{E}[T^{-1}]}{N - N_{\mathrm{trg}}} & (j \notin [N_{\mathrm{trg}}], k \in [N_{\mathrm{trg}}]) \\ \tilde{\eta}_V \frac{N_{\mathrm{trg}} + \mathbb{E}[T^{-1}](N(N-1) - N_{\mathrm{trg}}(N+2))}{(N - N_{\mathrm{trg}})^2} & (j = k \notin [N_{\mathrm{trg}}]) \\ \tilde{\eta}_V \frac{N_{\mathrm{trg}} - \mathbb{E}[T^{-1}](N + 2N_{\mathrm{trg}})}{(N - N_{\mathrm{trg}})^2} & (j \neq k, \, j, k \notin [N_{\mathrm{trg}}]) \end{cases} \qquad \text{(B.2)}$$

*and*

$$\langle e_j, W_V^{*,(3)} e_k \rangle = \begin{cases} -\tilde{\eta}_V \mathbb{E}[T^{-1}] N_{\mathrm{trg}}^{-1} & (j, k \in [N_{\mathrm{trg}}]) \\ -\tilde{\eta}_V \frac{1 - 2\mathbb{E}[T^{-1}]}{N - N_{\mathrm{trg}}} & (j \in [N_{\mathrm{trg}}], k \notin [N_{\mathrm{trg}}]) \\ \tilde{\eta}_V \frac{\mathbb{E}[T^{-1}]}{N - N_{\mathrm{trg}}} & (j \notin [N_{\mathrm{trg}}], k \in [N_{\mathrm{trg}}]) \\ \tilde{\eta}_V \frac{N_{\mathrm{trg}} + \mathbb{E}[T^{-1}](N(N-1) - N_{\mathrm{trg}}(N+2))}{(N - N_{\mathrm{trg}})^2} & (j = k \notin [N_{\mathrm{trg}}]) \\ \tilde{\eta}_V \frac{N_{\mathrm{trg}} - \mathbb{E}[T^{-1}](N + 2N_{\mathrm{trg}})}{(N - N_{\mathrm{trg}})^2} & (j \neq k, \, j, k \notin [N_{\mathrm{trg}}]), \end{cases}$$

*where we defined $\alpha_t := \langle \mathbb{E}_T[T^{-1} \mathbf{1}_{1:T}], p_t \rangle$: specifically, $\alpha_t = \mathbb{E}_T[T^{-1} \mathbb{I}\{t \leq T\}] \leq \mathbb{E}_T[T^{-1}]$ is satisfied.*

**Proof.** For the first block, we have the following evaluation of the gradient of population loss $\bar{\mathcal{L}}$:

$$\nabla_{W_V^{(1)}} \bar{\mathcal{L}}(W_{KQ}, W_V)$$
$$= \frac{1}{N} \sum_{k=1}^{N} e_k \mathbb{E}_T \left[ \frac{1}{T} \sum_{t=1}^{T} \mathbb{E}[p_t]^\top \right] - \frac{1}{N - N_{\mathrm{trg}}} \sum_{k=N_{\mathrm{trg}}+1}^{N} e_k \mathbb{E}_T \left[ \frac{1}{T} \sum_{t=1}^{T} \mathbb{E}[p_t | z_{T+1} = k]^\top \right]$$
$$= \frac{1}{N} \sum_{k=1}^{N} e_k \mathbb{E}_T \left[ T^{-1} \sum_{t=1}^{T} p_t^\top \right] - \frac{1}{N - N_{\mathrm{trg}}} \sum_{k=N_{\mathrm{trg}}+1}^{N} e_k \mathbb{E}_T \left[ T^{-1} \sum_{t=1}^{T} p_t^\top \right].$$

This immediately yields (B.1).

Now for the second block, we have

$$\nabla_{W_V^{(2)}} \bar{\mathcal{L}}(W_{KQ}, W_V)$$
$$= \frac{1}{N} \sum_{k=1}^{N} e_k \mathbb{E}_T \left[ \frac{1}{T} \sum_{t=1}^{T} \mathbb{E}[e_{z_t}]^\top \right] - \frac{1}{N - N_{\mathrm{trg}}} \sum_{k=N_{\mathrm{trg}}+1}^{N} e_k \mathbb{E}_T \left[ \frac{1}{T} \sum_{t=1}^{T} \mathbb{E}[e_{z_t} | z_{T+1} = k]^\top \right],$$

where the first term is evaluated as

$$\frac{1}{N} \sum_{k=1}^{N} e_k \mathbb{E}_T \left[ \frac{1}{T} \sum_{t=1}^{T} \mathbb{E}[e_{z_t}]^\top \right] = \frac{1}{N} \sum_{k=1}^{N} e_k \sum_{w=1}^{N_{\mathrm{trg}}} \frac{1}{N_{\mathrm{trg}}} \mathbb{E}_T \left[ \frac{1}{T} \left( 2e_w + \frac{T-2}{N - N_{\mathrm{trg}}} \mathbf{1}_{N_{\mathrm{trg}}+1:N} \right)^\top \right],$$

and similarly for the second term

$$\frac{1}{N - N_{\mathrm{trg}}} \sum_{k=N_{\mathrm{trg}}+1}^{N} e_k \mathbb{E}_T \left[ \frac{1}{T} \sum_{t=1}^{T} \mathbb{E}[e_{z_t} | z_{T+1} = k]^\top \right]$$
$$= \frac{1}{N - N_{\mathrm{trg}}} \sum_{k=N_{\mathrm{trg}}+1}^{N} e_k \frac{1}{N_{\mathrm{trg}}} \sum_{w=1}^{N_{\mathrm{trg}}} \mathbb{E}_T \left[ \frac{1}{T} \left( 2e_w + \frac{T-3}{N - N_{\mathrm{trg}}} \mathbf{1}_{N_{\mathrm{trg}}+1:N} + e_k \right)^\top \right].$$

Putting everything together yields (B.2).

The third block can be computed in the same fashion as

$$\nabla_{W_V^{(3)}} \bar{\mathcal{L}}(W_{KQ}, W_V)$$
$$= \frac{1}{N} \sum_{k=1}^{N} e_k \mathbb{E}_T \left[ \frac{1}{T} \sum_{t=1}^{T} \mathbb{E}[e_{z_{t-1}}]^\top \right] - \frac{1}{N - N_{\mathrm{trg}}} \sum_{k=N_{\mathrm{trg}}+1}^{N} e_k \mathbb{E}_T \left[ \frac{1}{T} \sum_{t=1}^{T} \mathbb{E}[e_{z_{t-1}} | z_{T+1} = k]^\top \right],$$

where we have the evaluations

$$\frac{1}{N}\sum_{k=1}^{N} \boldsymbol{e}_k \mathbb{E}_T\left[\frac{1}{T}\sum_{t=1}^{T}\mathbb{E}[\boldsymbol{e}_{z_{t-1}}]^\top\right]$$

$$=\frac{1}{N}\sum_{k=1}^{N} \boldsymbol{e}_k \frac{1}{N_{\text{trg}}}\sum_{w=1}^{N_{\text{trg}}}\mathbb{E}_T\left[\frac{1}{T}\left(\boldsymbol{e}_w + \frac{T-2}{N-N_{\text{trg}}}\mathbf{1}_{N_{\text{trg}}+1:N}\right)^\top\right] \quad (\because \boldsymbol{e}_0 = \mathbf{0})$$

and

$$\frac{1}{N-N_{\text{trg}}}\sum_{k=N_{\text{trg}}+1}^{N} \boldsymbol{e}_k \mathbb{E}_T\left[\frac{1}{T}\sum_{t=1}^{T}\mathbb{E}[\boldsymbol{e}_{z_{t-1}}|z_{T+1}=k]^\top\right]$$

$$=\frac{1}{N-N_{\text{trg}}}\sum_{k=N_{\text{trg}}+1}^{N} \boldsymbol{e}_k \frac{1}{N_{\text{trg}}}\sum_{w=1}^{N_{\text{trg}}}\mathbb{E}_T\left[\frac{1}{T}\left(\boldsymbol{e}_w + \frac{T-3}{N-N_{\text{trg}}}\mathbf{1}_{N_{\text{trg}}+1:N} + \boldsymbol{e}_k\right)^\top\right].$$

$\square$

## B.2  Population Gradient of $\mathbf{W}_{KQ}$

### B.2.1  Preparations

We denote the transformer's predicted probability of token $k$ given an input sequence $z$ after one-step GD on $\boldsymbol{W}_V$ as

$$\hat{p}(k|z) = \text{Softmax}(f_{\text{TF}}(\boldsymbol{X}_{1:T}; \boldsymbol{W}_{KQ}, \boldsymbol{W}_V^*))_k.$$

We can approximate $\hat{p}(k|z)$ by considering sufficiently small $\tilde{\eta}_V$: The following corollary is obtained by Lemma 9.

**Corollary 10.** *If we set $\tilde{\eta}_V \lesssim 1/N$, then $|1/N - \hat{p}(k|z)| = O(\mathbb{E}[T^{-1}]/N^2)$ uniformly holds for any $k$ and $z$.*

**Proof.** From Lemma 9, it holds that $|(f_{\text{TF}}(\boldsymbol{X}_{1:T}; \boldsymbol{W}_{KQ}, \boldsymbol{W}_V^*))_k| \in O(\mathbb{E}[T^{-1}]/N + 1/N^2) \in O(\mathbb{E}[T^{-1}]/N)$ ($\because$ Assumption 1) for all $k$. If $v_i = O(\mathbb{E}[T^{-1}]/N)$ for all $i \in [N]$ where $\boldsymbol{v} \in \mathbb{R}^N$, then it holds that

$$\frac{1}{N}\exp\big(-O(\mathbb{E}[T^{-1}]/N)\big) \le \text{Softmax}(\boldsymbol{v})_i \le \frac{1}{N}\exp\big(O(\mathbb{E}[T^{-1}]/N)\big).$$

From Taylor's theorem we obtain the assertion. $\square$

Following [BCB$^+$23, Lemma 4], starting from $\boldsymbol{W}_{KQ} = \mathbf{0}$,

$$\nabla_{\boldsymbol{W}_{KQ}}\bar{\mathcal{L}}(\boldsymbol{W}_{KQ}, \boldsymbol{W}_V^*)$$

$$=\mathbb{E}\left[\sum_{k=1}^{N}(\hat{p}(k|z) - \mathbb{I}\{z_{T+1}=k\})\nabla_{\boldsymbol{W}_{KQ}=\mathbf{0}}\big\langle \boldsymbol{e}_k, \boldsymbol{W}_V^* \boldsymbol{X}_{1:T}\text{Softmax}(\boldsymbol{X}_{1:T}^\top \boldsymbol{W}_{KQ}\boldsymbol{x}_T)\big\rangle\right],$$

where

$$\nabla_{\boldsymbol{W}_{KQ}=\mathbf{0}}\left(\boldsymbol{e}_k^\top \boldsymbol{W}_V^* \sum_{t=1}^{T} \boldsymbol{x}_t\text{Softmax}(\boldsymbol{X}_{1:T}^\top \boldsymbol{W}_{KQ}\boldsymbol{x}_T)_t\right)$$

$$=\sum_{t=1}^{T}\boldsymbol{e}_k^\top \boldsymbol{W}_V^* \boldsymbol{x}_t \cdot \nabla_{\boldsymbol{W}_{KQ}=\mathbf{0}}\text{Softmax}(\boldsymbol{X}_{1:T}^\top \boldsymbol{W}_{KQ}\boldsymbol{x}_T)_t$$

$$=\frac{1}{T}\sum_{t=1}^{T}\boldsymbol{e}_k^\top \boldsymbol{W}_V^* \boldsymbol{x}_t \cdot (\boldsymbol{x}_t - \bar{\boldsymbol{x}}_{1:T})\boldsymbol{x}_T^\top,$$

for $\bar{\boldsymbol{x}}_{1:T} = \frac{1}{T}\sum_{t=1}^{T}\boldsymbol{x}_t$. Hence the population gradient simplifies to, assuming $\tilde{\eta}_V \lesssim 1/N$,

$$\nabla_{\boldsymbol{W}_{KQ}}\bar{\mathcal{L}}(\boldsymbol{W}_{KQ}, \boldsymbol{W}_V^*)$$

$$= \sum_{k=1}^{N} \mathbb{E}_T \left[ \frac{1}{T} \sum_{t=1}^{T} \mathbb{E}_{\boldsymbol{X}} \left[ \hat{p}(k|z) \boldsymbol{e}_k^\top \boldsymbol{W}_V^* \boldsymbol{x}_t \cdot (\boldsymbol{x}_t - \bar{\boldsymbol{x}}_{1:T}) \boldsymbol{x}_T^\top \right] \right]$$

$$- \sum_{k=N_{\text{trg}}+1}^{N} \mathbb{E}_T \left[ \frac{1}{T} \sum_{t=1}^{T} p(z_{T+1} = k) \mathbb{E}_{\boldsymbol{X}} \left[ \boldsymbol{e}_k^\top \boldsymbol{W}_V^* \boldsymbol{x}_t \cdot (\boldsymbol{x}_t - \bar{\boldsymbol{x}}_{1:T}) \boldsymbol{x}_T^\top | z_{T+1} = k \right] \right]$$

$$= \frac{1}{N} \sum_{k=1}^{N} \mathbb{E}_T \left[ \frac{1}{T} \sum_{t=1}^{T} \mathbb{E}_{\boldsymbol{X}} \left[ \boldsymbol{e}_k^\top \boldsymbol{W}_V^* \boldsymbol{x}_t \cdot (\boldsymbol{x}_t - \bar{\boldsymbol{x}}_{1:T}) \boldsymbol{x}_T^\top \right] \right]$$

$$- \frac{1}{N - N_{\text{trg}}} \sum_{k=N_{\text{trg}}+1}^{N} \mathbb{E}_T \left[ \frac{1}{T} \sum_{t=1}^{T} \mathbb{E}_{\boldsymbol{X}} \left[ \boldsymbol{e}_k^\top \boldsymbol{W}_V^* \boldsymbol{x}_t \cdot (\boldsymbol{x}_t - \bar{\boldsymbol{x}}_{1:T}) \boldsymbol{x}_T^\top | z_{T+1} = k \right] \right]$$

$$+ \underbrace{\sum_{k=1}^{N} \mathbb{E}_T \left[ \frac{1}{T} \sum_{t=1}^{T} \mathbb{E}_{\boldsymbol{X}} \left[ \left( \hat{p}(k|z) - \frac{1}{N} \right) \boldsymbol{e}_k^\top \boldsymbol{W}_V^* \boldsymbol{x}_t \cdot (\boldsymbol{x}_t - \bar{\boldsymbol{x}}_{1:T}) \boldsymbol{x}_T^\top \right] \right]}_{(*1)}.$$

Note that each entry of $\boldsymbol{e}_k^\top \boldsymbol{W}_V^* \boldsymbol{x}_t \cdot (\boldsymbol{x}_t - \bar{\boldsymbol{x}}_{1:T}) \boldsymbol{x}_T^\top$ is of $O(\tilde{\eta}_V \mathbb{E}[T^{-1}])$ from Lemma 9, and $|\hat{p}(k|z) - \frac{1}{N}| \lesssim \mathbb{E}[T^{-1}]/N^2$ from Corollary 10. Therefore, using $|\mathbb{E}[ab]| \leq \sqrt{\mathbb{E}[a^2 b^2]}$ we can conclude that $(*1)$ is of $O_\infty(\tilde{\eta}_V \mathbb{E}[T^{-1}]^2/N) \in O_\infty(\tilde{\eta}_V \mathbb{E}[T^{-1}] N_{\text{trg}}/N)$.

Moreover, we have

$$\frac{1}{N} \sum_{k=1}^{N} \mathbb{E}_T \left[ \frac{1}{T} \sum_{t=1}^{T} \mathbb{E}_{\boldsymbol{X}} \left[ \boldsymbol{e}_k^\top \boldsymbol{W}_V^* \boldsymbol{x}_t \cdot (\boldsymbol{x}_t - \bar{\boldsymbol{x}}_{1:T}) \boldsymbol{x}_T^\top \right] \right]$$

$$= \frac{1}{N - N_{\text{trg}}} \sum_{k=N_{\text{trg}}+1}^{N} \mathbb{E}_T \left[ \frac{1}{T} \sum_{t=1}^{T} \mathbb{E}_{\boldsymbol{X}} \left[ \boldsymbol{e}_k^\top \boldsymbol{W}_V^* \boldsymbol{x}_t \cdot (\boldsymbol{x}_t - \bar{\boldsymbol{x}}_{1:T}) \boldsymbol{x}_T^\top \right] \right]$$

$$+ \frac{1}{N - N_{\text{trg}}} \sum_{k=1}^{N_{\text{trg}}} \mathbb{E}_T \left[ \frac{1}{T} \sum_{t=1}^{T} \mathbb{E}_{\boldsymbol{X}} \left[ \boldsymbol{e}_k^\top \boldsymbol{W}_V^* \boldsymbol{x}_t \cdot (\boldsymbol{x}_t - \bar{\boldsymbol{x}}_{1:T}) \boldsymbol{x}_T^\top \right] \right]$$

$$- \frac{N_{\text{trg}}}{N(N - N_{\text{trg}})} \sum_{k=1}^{N} \mathbb{E}_T \left[ \frac{1}{T} \sum_{t=1}^{T} \mathbb{E}_{\boldsymbol{X}} \left[ \boldsymbol{e}_k^\top \boldsymbol{W}_V^* \boldsymbol{x}_t \cdot (\boldsymbol{x}_t - \bar{\boldsymbol{x}}_{1:T}) \boldsymbol{x}_T^\top \right] \right].$$

Here, again using the fact that each entry of $\boldsymbol{e}_k^\top \boldsymbol{W}_V^* \boldsymbol{x}_t \cdot (\boldsymbol{x}_t - \bar{\boldsymbol{x}}_{1:T}) \boldsymbol{x}_T^\top$ is of $O(\tilde{\eta}_V \mathbb{E}[T^{-1}])$, the second and third terms can be bounded by $O_\infty(\tilde{\eta}_V \mathbb{E}[T^{-1}] \cdot N_{\text{trg}}/N)$.

In conclusion, we obtain

$$\nabla_{\boldsymbol{W}_{KQ}} \bar{\mathcal{L}}(\boldsymbol{W}_{KQ}, \boldsymbol{W}_V^*)$$

$$= \frac{1}{N - N_{\text{trg}}} \sum_{k=N_{\text{trg}}+1}^{N} \mathbb{E}_T \left[ \frac{1}{T} \sum_{t=1}^{T} \mathbb{E}_{\boldsymbol{X}} \left[ \boldsymbol{e}_k^\top \boldsymbol{W}_V^* \boldsymbol{x}_t \cdot (\boldsymbol{x}_t - \bar{\boldsymbol{x}}_{1:T}) \boldsymbol{x}_T^\top \right] \right]$$

$$- \frac{1}{N - N_{\text{trg}}} \sum_{k=N_{\text{trg}}+1}^{N} \mathbb{E}_T \left[ \frac{1}{T} \sum_{t=1}^{T} \mathbb{E}_{\boldsymbol{X}} \left[ \boldsymbol{e}_k^\top \boldsymbol{W}_V^* \boldsymbol{x}_t \cdot (\boldsymbol{x}_t - \bar{\boldsymbol{x}}_{1:T}) \boldsymbol{x}_T^\top | z_{T+1} = k \right] \right]$$

$$+ O_\infty(\tilde{\eta}_V \mathbb{E}[T^{-1}] \cdot N_{\text{trg}}/N).$$

### B.2.2 Detailed Calculations

Let

$$\boldsymbol{\Delta}(k, T)$$

$$:= \frac{1}{T} \sum_{t=1}^{T} \left( \mathbb{E}_{\boldsymbol{X}} \left[ \boldsymbol{e}_k^\top \boldsymbol{W}_V^* \boldsymbol{x}_t \cdot (\boldsymbol{x}_t - \bar{\boldsymbol{x}}_{1:T}) \boldsymbol{x}_T^\top \right] - \mathbb{E}_{\boldsymbol{X}} \left[ \boldsymbol{e}_k^\top \boldsymbol{W}_V^* \boldsymbol{x}_t \cdot (\boldsymbol{x}_t - \bar{\boldsymbol{x}}_{1:T}) \boldsymbol{x}_T^\top | z_{T+1} = k \right] \right)$$

for $N_{\text{trg}} + 1 \leq k \leq N$. Then, it holds that $\nabla_{\boldsymbol{W}_{KQ}} \bar{\mathcal{L}}(\boldsymbol{W}_{KQ}, \boldsymbol{W}_V^*) = \frac{1}{N-N_{\text{trg}}} \sum_{k=N_{\text{trg}}+1}^{N} \mathbb{E}_T[\boldsymbol{\Delta}(k,T)] + \tilde{\eta}_V \mathbb{E}[T^{-1}] O_\infty(N_{\text{trg}}/N)$.

For the first and second (block) columns, we have the following lemma:

**Lemma 11.** *Let $\boldsymbol{W}_{KQ}^* = -\eta_{KQ} \nabla_{\boldsymbol{W}_{KQ}} \bar{\mathcal{L}}(\boldsymbol{W}_{KQ}, \boldsymbol{W}_V^*)$. Then, it holds that*

$$
\boldsymbol{W}_{KQ}^*[:, 1 : L + N]
$$

$$
= \tilde{\eta}_V \eta_{KQ} \frac{1}{N_{\text{trg}}} \sum_{w=1}^{N_{\text{trg}}} \sum_{\ell} p(\ell) \left( T(\ell)^{-1} \begin{bmatrix} \boldsymbol{p}_{\ell+2} \mathbb{E}_{\ell'}[T(\ell')^{-1}] + \boldsymbol{p}_{\ell+3} \mathbb{E}_{\ell'}[T(\ell')^{-1}] \\ \boldsymbol{0} \\ \boldsymbol{e}_w \mathbb{E}_{\ell'}[T(\ell')^{-1}] \end{bmatrix} \begin{bmatrix} \boldsymbol{p}_{T(\ell)}^\top & \boldsymbol{e}_w^\top \end{bmatrix} \right.
$$

$$
\left. - 2T(\ell)^{-2} \begin{bmatrix} \boldsymbol{1}_{1:T(\ell) \setminus \{\ell+2, \ell+3\}} \mathbb{E}_{\ell'}[T(\ell')^{-1}] \\ 2\boldsymbol{e}_w \mathbb{E}_{\ell'}[T(\ell')^{-1}] \\ \boldsymbol{0} \end{bmatrix} \begin{bmatrix} \boldsymbol{p}_{T(\ell)}^\top & \boldsymbol{e}_w^\top \end{bmatrix} \right)
$$

$$
+ \tilde{\eta}_V \eta_{KQ} \mathbb{E}_{\ell'}[T(\ell')^{-1}] O(N_{\text{trg}} \cdot N^{-1}).
$$

*Here $O(N_{\text{trg}} \cdot N^{-1})$ denotes a matrix whose entries are all of $O(N_{\text{trg}} \cdot N^{-1})$ and $p(\ell)$ is the probability of drawing $\ell$ at pretraining data.*

**Proof.** [Proof of Lemma 11] Suppose $\boldsymbol{W}_V$ is obtained by Lemma 9. Note that

$$
\boldsymbol{\Delta}(k, T)[:, : L]
$$

$$
= \frac{1}{T} \sum_{t=1}^{T} \left( \mathbb{E}_{\boldsymbol{X}} \left[ \boldsymbol{e}_k^\top \boldsymbol{W}_V \boldsymbol{x}_t \cdot (\boldsymbol{x}_t - \bar{\boldsymbol{x}}_{1:T}) \right] - \mathbb{E}_{\boldsymbol{X}} \left[ \boldsymbol{e}_k^\top \boldsymbol{W}_V \boldsymbol{x}_t \cdot (\boldsymbol{x}_t - \bar{\boldsymbol{x}}_{1:T}) | z_{T+1} = k \right] \right) \boldsymbol{p}_T^\top
$$

$$
= \frac{1}{N_{\text{trg}}} \sum_{w=1}^{N_{\text{trg}}} \frac{1}{T} \sum_{t=1}^{T} \left( \mathbb{E}_{\boldsymbol{X}} \left[ (\boldsymbol{x}_t - \bar{\boldsymbol{x}}_{1:T}) \boldsymbol{x}_t^\top | \boldsymbol{x}_T = w \right] - \mathbb{E}_{\boldsymbol{X}} \left[ (\boldsymbol{x}_t - \bar{\boldsymbol{x}}_{1:T}) \boldsymbol{x}_t^\top | z_{T+1} = k, \boldsymbol{x}_T = w \right] \right)
$$

$$
\cdot \boldsymbol{W}_V^\top \boldsymbol{e}_k \boldsymbol{p}_T^\top
$$

and similarly

$$
\boldsymbol{\Delta}(k, T)[:, L + 1 : L + N]
$$

$$
= \frac{1}{N_{\text{trg}}} \sum_{w=1}^{N_{\text{trg}}} \left[ \frac{1}{T} \sum_{t=1}^{T} \left( \mathbb{E}_{\boldsymbol{X}} \left[ (\boldsymbol{x}_t - \bar{\boldsymbol{x}}_{1:T}) \boldsymbol{x}_t^\top | \boldsymbol{x}_T = w \right] - \mathbb{E}_{\boldsymbol{X}} \left[ (\boldsymbol{x}_t - \bar{\boldsymbol{x}}_{1:T}) \boldsymbol{x}_t^\top | z_{T+1} = k, \boldsymbol{x}_T = w \right] \right) \right.
$$

$$
\left. \cdot \boldsymbol{W}_V^\top \boldsymbol{e}_k \boldsymbol{e}_w^\top \right].
$$

Now let us consider the difference

$$
\frac{1}{T} \sum_{t=1}^{T} \left( \mathbb{E}_{\boldsymbol{X}} \left[ (\boldsymbol{x}_t - \bar{\boldsymbol{x}}_{1:T}) \boldsymbol{x}_t^\top | \boldsymbol{x}_T = w \right] - \mathbb{E}_{\boldsymbol{X}} \left[ (\boldsymbol{x}_t - \bar{\boldsymbol{x}}_{1:T}) \boldsymbol{x}_t^\top | z_{T+1} = k, \boldsymbol{x}_T = w \right] \right)
$$

$$
= \frac{1}{T} \sum_{t=1}^{T} \left( \mathbb{E}_{\boldsymbol{X}} \left[ \boldsymbol{x}_t \boldsymbol{x}_t^\top | \boldsymbol{x}_T = w \right] - \mathbb{E}_{\boldsymbol{X}} \left[ \boldsymbol{x}_t \boldsymbol{x}_t^\top | z_{T+1} = k, \boldsymbol{x}_T = w \right] \right)
$$

$$
- \frac{1}{T^2} \sum_{t=1}^{T} \sum_{t'=1}^{T} \left( \mathbb{E}_{\boldsymbol{X}} \left[ \boldsymbol{x}_t \boldsymbol{x}_{t'}^\top | \boldsymbol{x}_T = w \right] - \mathbb{E}_{\boldsymbol{X}} \left[ \boldsymbol{x}_t \boldsymbol{x}_{t'}^\top | z_{T+1} = k, \boldsymbol{x}_T = w \right] \right)
$$

$$
= \left( \frac{1}{T} - \frac{1}{T^2} \right) \sum_{t=1}^{T} \left( \mathbb{E}_{\boldsymbol{X}} \left[ \boldsymbol{x}_t \boldsymbol{x}_t^\top | \boldsymbol{x}_T = w \right] - \mathbb{E}_{\boldsymbol{X}} \left[ \boldsymbol{x}_t \boldsymbol{x}_t^\top | z_{T+1} = k, \boldsymbol{x}_T = w \right] \right)
$$

$$
- \frac{1}{T^2} \sum_{t \neq t'} \left( \mathbb{E}_{\boldsymbol{X}} \left[ \boldsymbol{x}_t \boldsymbol{x}_{t'}^\top | \boldsymbol{x}_T = w \right] - \mathbb{E}_{\boldsymbol{X}} \left[ \boldsymbol{x}_t \boldsymbol{x}_{t'}^\top | z_{T+1} = k, \boldsymbol{x}_T = w \right] \right)
$$

Then it suffices to calculate the vector

$$
\boldsymbol{d}(t,t',k,w) = \boldsymbol{M}(t,t',k,w)\boldsymbol{W}_V^\top \boldsymbol{e}_k
$$
$$
:= \left(\mathbb{E}[\boldsymbol{x}_t \boldsymbol{x}_{t'}^\top | \boldsymbol{x}_T = w] - \mathbb{E}[\boldsymbol{x}_t \boldsymbol{x}_{t'}^\top | z_{T+1} = k, \boldsymbol{x}_T = w]\right)\boldsymbol{W}_V^\top \boldsymbol{e}_k
$$

for each $t, t', k$ and $w$. Recall

$$
\boldsymbol{W}_V^\top \boldsymbol{e}_k = \tilde{\eta_V}
\begin{bmatrix}
\frac{\alpha_1 N_{\text{trg}}}{N - N_{\text{trg}}} \\
\vdots \\
\frac{\alpha_L N_{\text{trg}}}{N - N_{\text{trg}}} \\
\hline
\frac{2}{N - N_{\text{trg}}}\mathbb{E}[T^{-1}]\mathbf{1}_{N_{\text{trg}}} \\
(-\frac{1}{N}\mathbb{E}[T^{-1}] + O(N_{\text{trg}} \cdot N^{-2}))\mathbf{1}_{k - N_{\text{trg}} - 1} \\
\mathbb{E}[T^{-1}] + \mathbb{E}[T^{-1}]O(N_{\text{trg}} \cdot N^{-1}) + O(N_{\text{trg}} \cdot N^{-2}) \\
(-\frac{1}{N}\mathbb{E}[T^{-1}] + O(N_{\text{trg}} \cdot N^{-2}))\mathbf{1}_{N-k} \\
\hline
\frac{1}{N - N_{\text{trg}}}\mathbb{E}[T^{-1}]\mathbf{1}_{N_{\text{trg}}} \\
(-\frac{1}{N}\mathbb{E}[T^{-1}] + O(N_{\text{trg}} \cdot N^{-2}))\mathbf{1}_{k - N_{\text{trg}} - 1} \\
\mathbb{E}[T^{-1}] + \mathbb{E}[T^{-1}]O(N_{\text{trg}} \cdot N^{-1}) + O(N_{\text{trg}} \cdot N^{-2}) \\
(-\frac{1}{N}\mathbb{E}[T^{-1}] + O(N_{\text{trg}} \cdot N^{-2}))\mathbf{1}_{N-k}
\end{bmatrix}.
$$

For preparation, we define some vectors: let

$$
\boldsymbol{\alpha}(k) = \left[\underbrace{0, 0, \ldots, 0}_{N_{\text{trg}} \text{ zeros}}, \frac{1}{N - N_{\text{trg}}}, \ldots, \frac{1}{N - N_{\text{trg}}}, \underbrace{\frac{N + N_{\text{trg}} + 1}{N - N_{\text{trg}}}}_{k\text{-th entry}}, \frac{1}{N - N_{\text{trg}}}, \ldots, \frac{1}{N - N_{\text{trg}}}\right]^\top
$$

be an $N$-dimensional vector for $N_{\text{trg}} + 1 \leq k \leq N$ and

$$
\boldsymbol{\beta} = \left[\underbrace{0, 0, \ldots, 0}_{N_{\text{trg}} \text{ zeros}}, \frac{1}{N - N_{\text{trg}}}, \ldots, \frac{1}{N - N_{\text{trg}}}\right]^\top \in \mathbb{R}^N.
$$

First, if $t = t'$, then $\boldsymbol{M}(t,t',k,w)$ is zero unless $t = \ell + 2$ or $t = \ell + 3$, as $z_{T+1}$ is independent of $z_i$ $(i \in [T], i \neq \ell + 2)$ and only $\boldsymbol{x}_{\ell+2}$ and $\boldsymbol{x}_{\ell+3}$ include the information of $z_{\ell+2}$. For each case, we have

$$
\boldsymbol{M}(\ell + 2, \ell + 2, k, w) = \begin{bmatrix}
\boldsymbol{O}_{L \times L} & \boldsymbol{p}_{\ell+2}\boldsymbol{\alpha}(k)^\top & \boldsymbol{O}_{L \times N} \\
\boldsymbol{\alpha}(k)\boldsymbol{p}_{\ell+2}^\top & \text{diag}(\boldsymbol{\alpha}(k)) & \boldsymbol{\alpha}(k)\boldsymbol{e}_w^\top \\
\boldsymbol{O}_{N \times L} & \boldsymbol{e}_w\boldsymbol{\alpha}(k)^\top & \boldsymbol{O}_{N \times N}
\end{bmatrix}
$$

and

$$
\boldsymbol{M}(\ell + 3, \ell + 3, k, w) = \begin{bmatrix}
\boldsymbol{O}_{L \times L} & \boldsymbol{O}_{L \times N} & \boldsymbol{p}_{\ell+3}\boldsymbol{\alpha}(k)^\top \\
\boldsymbol{O}_{N \times L} & \boldsymbol{O}_{N \times N} & \boldsymbol{\beta}\boldsymbol{\alpha}(k)^\top \\
\boldsymbol{\alpha}(k)\boldsymbol{p}_{\ell+3}^\top & \boldsymbol{\alpha}(k)\boldsymbol{\beta}^\top & \text{diag}(\boldsymbol{\alpha}(k))
\end{bmatrix},
$$

then we obtain

$$
\boldsymbol{d}(\ell + 2, \ell + 2, k, w) = \tilde{\eta_V}\begin{bmatrix}
-\boldsymbol{p}_{\ell+2}\mathbb{E}[T^{-1}] \\
-\boldsymbol{e}_k\mathbb{E}[T^{-1}] \\
-\boldsymbol{e}_w\mathbb{E}[T^{-1}]
\end{bmatrix} + \tilde{\eta_V}O_\infty(\mathbb{E}[T^{-1}]\frac{N_{\text{trg}}}{N} + \frac{N_{\text{trg}}}{N^2})
$$

and

$$
\boldsymbol{d}(\ell + 3, \ell + 3, k, w) = \tilde{\eta_V}\begin{bmatrix}
-\boldsymbol{p}_{\ell+3}\mathbb{E}[T^{-1}] \\
\mathbf{0} \\
-\boldsymbol{e}_k\mathbb{E}[T^{-1}]
\end{bmatrix} + \tilde{\eta_V}O_\infty(\mathbb{E}[T^{-1}]\frac{N_{\text{trg}}}{N} + \frac{N_{\text{trg}}}{N^2}).
$$

For the case $t \neq t'$, deal with the following three cases:

(i) If $(t,t') = (\ell+2, \ell+3)$ or $(t,t') = (\ell+3, \ell+2)$ then we have

$$\boldsymbol{M}(\ell+2, \ell+3, k, w) + \boldsymbol{M}(\ell+3, \ell+2, k, w)$$
$$= \begin{bmatrix} \boldsymbol{O}_{L\times L} & \boldsymbol{p}_{\ell+3}\boldsymbol{\alpha}(k)^\top & \boldsymbol{p}_{\ell+2}\boldsymbol{\alpha}(k)^\top \\ \boldsymbol{\alpha}(k)\boldsymbol{p}_{\ell+3}^\top & \boldsymbol{\alpha}(k)\boldsymbol{\beta}^\top + \boldsymbol{\beta}\boldsymbol{\alpha}(k)^\top & \mathrm{diag}(\boldsymbol{\alpha}(k)) \\ \boldsymbol{\alpha}(k)\boldsymbol{p}_{\ell+2}^\top & \mathrm{diag}(\boldsymbol{\alpha}(k)) & \boldsymbol{\alpha}(k)\boldsymbol{e}_w^\top + \boldsymbol{e}_w\boldsymbol{\alpha}(k)^\top \end{bmatrix}$$

and

$$\boldsymbol{d}(\ell+2, \ell+3, k, w) + \boldsymbol{d}(\ell+3, \ell+2, k, w)$$
$$= \tilde{\eta}_V \begin{bmatrix} -\boldsymbol{p}_{\ell+2}\mathbb{E}[T^{-1}] - \boldsymbol{p}_{\ell+3}\mathbb{E}[T^{-1}] \\ -\boldsymbol{e}_k\mathbb{E}[T^{-1}] \\ -\boldsymbol{e}_k\mathbb{E}[T^{-1}] - \boldsymbol{e}_w\mathbb{E}[T^{-1}] \end{bmatrix} + \tilde{\eta}_V O_\infty (\mathbb{E}[T^{-1}]\frac{N_{\mathrm{trg}}}{N} + \frac{N_{\mathrm{trg}}}{N^2}).$$

(ii) For $t \neq \ell+2, \ell+3$ we have

$$\boldsymbol{M}(t, \ell+2, k, w) + \boldsymbol{M}(\ell+2, t, k, w) = \begin{bmatrix} \boldsymbol{O} & \boldsymbol{p}_t\boldsymbol{\alpha}(k)^\top & \boldsymbol{O} \\ \boldsymbol{\alpha}(k)\boldsymbol{p}_t^\top & \boldsymbol{\gamma}(t)\boldsymbol{\alpha}(k)^\top + \boldsymbol{\alpha}(k)\boldsymbol{\gamma}(t)^\top & \boldsymbol{\alpha}(k)\boldsymbol{\beta}^\top \\ \boldsymbol{O} & \boldsymbol{\beta}\boldsymbol{\alpha}(k)^\top & \boldsymbol{O} \end{bmatrix}$$

where $\boldsymbol{\gamma}(t) = \boldsymbol{e}_w$ if $t = \ell+1$ or $t = T$ and $\boldsymbol{\gamma}(t) = \boldsymbol{\beta}$ otherwise. To summarize,

$$\boldsymbol{d}(t, \ell+2, k, w) + \boldsymbol{d}(\ell+2, t, k, w) = \tilde{\eta}_V \begin{bmatrix} -\boldsymbol{p}_t\mathbb{E}[T^{-1}] \\ -\boldsymbol{e}_w\mathbb{E}[T^{-1}] \\ \boldsymbol{0} \end{bmatrix} + \tilde{\eta}_V O_\infty (\mathbb{E}[T^{-1}]\frac{N_{\mathrm{trg}}}{N} + \frac{N_{\mathrm{trg}}}{N^2}).$$

if $t = \ell+1$ or $t = T$ and

$$\boldsymbol{d}(t, \ell+2, k, w) + \boldsymbol{d}(\ell+2, t, k, w) = \tilde{\eta}_V \begin{bmatrix} -\boldsymbol{p}_t\mathbb{E}[T^{-1}] \\ \boldsymbol{0} \\ \boldsymbol{0} \end{bmatrix} + \tilde{\eta}_V O_\infty (\mathbb{E}[T^{-1}]\frac{N_{\mathrm{trg}}}{N} + \frac{N_{\mathrm{trg}}}{N^2}).$$

otherwise.

(iii) For $t \neq \ell+2, \ell+3$ we have

$$\boldsymbol{M}(t, \ell+3, k, w) + \boldsymbol{M}(\ell+3, t, k, w) = \begin{bmatrix} \boldsymbol{O} & \boldsymbol{O} & \boldsymbol{p}_t\boldsymbol{\alpha}(k)^\top \\ \boldsymbol{O} & \boldsymbol{O} & \boldsymbol{\gamma}(t)\boldsymbol{\alpha}(k)^\top \\ \boldsymbol{\alpha}(k)\boldsymbol{p}_t^\top & \boldsymbol{\alpha}(k)\boldsymbol{\gamma}(t)^\top & \boldsymbol{\beta}\boldsymbol{\alpha}(k)^\top + \boldsymbol{\alpha}(k)\boldsymbol{\beta}^\top \end{bmatrix}$$

and we obtain

$$\boldsymbol{d}(t, \ell+3, k, w) + \boldsymbol{d}(\ell+3, t, k, w) = \tilde{\eta}_V \begin{bmatrix} -\boldsymbol{p}_t\mathbb{E}[T^{-1}] \\ -\boldsymbol{e}_w\mathbb{E}[T^{-1}] \\ \boldsymbol{0} \end{bmatrix} + \tilde{\eta}_V O_\infty (\mathbb{E}[T^{-1}]\frac{N_{\mathrm{trg}}}{N} + \frac{N_{\mathrm{trg}}}{N^2})$$

if $t = \ell+1$ or $t = T$ and

$$\boldsymbol{d}(t, \ell+3, k, w) + \boldsymbol{d}(\ell+3, t, k, w) = \tilde{\eta}_V \begin{bmatrix} -\boldsymbol{p}_t\mathbb{E}[T^{-1}] \\ \boldsymbol{0} \\ \boldsymbol{0} \end{bmatrix} + \tilde{\eta}_V O_\infty (\mathbb{E}[T^{-1}]\frac{N_{\mathrm{trg}}}{N} + \frac{N_{\mathrm{trg}}}{N^2}).$$

otherwise.

Now we are ready to calculate $-\frac{1}{N-N_{\mathrm{trg}}}\sum_{k=N_{\mathrm{trg}}+1}^N \boldsymbol{\Delta}(k,T)[:, : L + N]$ as

$$-\frac{1}{N-N_{\mathrm{trg}}}\sum_{k=N_{\mathrm{trg}}+1}^N \boldsymbol{\Delta}(k,T)[:, 1 : L + N]$$

$$= \frac{1}{N_{\mathrm{trg}}}\sum_{w=1}^{N_{\mathrm{trg}}}\left\{ -\frac{\tilde{\eta}_V}{N-N_{\mathrm{trg}}}\sum_{k=N_{\mathrm{trg}}+1}^N \left[ \left(\frac{1}{T} - \frac{1}{T^2}\right)\left( \begin{bmatrix} -\boldsymbol{p}_{\ell+2}\mathbb{E}[T^{-1}] \\ -\boldsymbol{e}_k\mathbb{E}[T^{-1}] \\ -\boldsymbol{e}_w\mathbb{E}[T^{-1}] \end{bmatrix} + \begin{bmatrix} -\boldsymbol{p}_{\ell+3}\mathbb{E}[T^{-1}] \\ \boldsymbol{0} \\ -\boldsymbol{e}_k\mathbb{E}[T^{-1}] \end{bmatrix} \right) \right. \right.$$

$$\left. \left. + \frac{1}{T^2}\begin{bmatrix} \boldsymbol{p}_{\ell+2}\mathbb{E}[T^{-1}] + \boldsymbol{p}_{\ell+3}\mathbb{E}[T^{-1}] \\ \boldsymbol{e}_k\mathbb{E}[T^{-1}] \\ \boldsymbol{e}_k\mathbb{E}[T^{-1}] + \boldsymbol{e}_w\mathbb{E}[T^{-1}] \end{bmatrix} + \frac{2}{T^2}\begin{bmatrix} \mathbf{1}_{1:T\setminus\{\ell+2,\ell+3\}}\mathbb{E}[T^{-1}] \\ 2\boldsymbol{e}_w\mathbb{E}[T^{-1}] \\ \boldsymbol{0} \end{bmatrix} \right]\begin{bmatrix} \boldsymbol{p}_T^\top & \boldsymbol{e}_w^\top \end{bmatrix} \right\}$$

$$+ \tilde{\eta}_V \mathbb{E}[T^{-1}] O(N_{\text{trg}} \cdot N^{-1}) + \tilde{\eta}_V O(N_{\text{trg}} \cdot N^{-2})$$

$$= \frac{1}{N_{\text{trg}}} \sum_{w=1}^{N_{\text{trg}}} \left\{ \frac{\tilde{\eta}_V}{T} \begin{bmatrix} \boldsymbol{p}_{\ell+2}\mathbb{E}[T^{-1}] + \boldsymbol{p}_{\ell+3}\mathbb{E}[T^{-1}] \\ \mathbf{0} \\ \boldsymbol{e}_w \mathbb{E}[T^{-1}] \end{bmatrix} \begin{bmatrix} \boldsymbol{p}_T^\top & \boldsymbol{e}_w^\top \end{bmatrix} \right.$$

$$\left. - \frac{2\tilde{\eta}_V}{T^2} \begin{bmatrix} \mathbf{1}_{1:T \setminus \{\ell+2,\ell+3\}}\mathbb{E}[T^{-1}] \\ 2\boldsymbol{e}_w\mathbb{E}[T^{-1}] \\ \mathbf{0} \end{bmatrix} \begin{bmatrix} \boldsymbol{p}_T^\top & \boldsymbol{e}_w^\top \end{bmatrix} \right\}$$

$$+ \tilde{\eta}_V \mathbb{E}[T^{-1}] O_\infty(N_{\text{trg}} \cdot N^{-1}) + \tilde{\eta}_V O_\infty(N_{\text{trg}} \cdot N^{-2}),$$

which concludes the proof together with $\mathbb{E}[T^{-1}] \gtrsim N^{-1}$ from Assumption 1. $\qquad \square$

We can also bound the last column:

**Lemma 12.** *It holds that*

$$\boldsymbol{W}_{KQ}^*[:, L+N+1:] = -\frac{1}{N - N_{\text{trg}}} \sum_{k=N_{\text{trg}}+1}^{N} \boldsymbol{\Delta}(k,T)[:, L+N+1:]$$

$$= \tilde{\eta}_V \eta_{KQ} \mathbb{E}_{\ell'}[T(\ell')^{-1}] O_\infty(N_{\text{trg}} \cdot N^{-1}).$$

**Proof.** Note that

$$\boldsymbol{\Delta}(k,T)[:, L+N+1:]$$

$$= \frac{1}{T} \sum_{t=1}^{T} \left( \mathbb{E}_{\boldsymbol{X}}\left[ \boldsymbol{e}_k^\top \boldsymbol{W}_V^* \boldsymbol{x}_t \cdot (\boldsymbol{x}_t - \bar{\boldsymbol{x}}_{1:T}) \boldsymbol{e}_{z_{T-1}}^\top \right] - \mathbb{E}_{\boldsymbol{X}}\left[ \boldsymbol{e}_k^\top \boldsymbol{W}_V^* \boldsymbol{x}_t \cdot (\boldsymbol{x}_t - \bar{\boldsymbol{x}}_{1:T}) \boldsymbol{e}_{z_{T-1}}^\top | z_{T+1} = k \right] \right)$$

$$= \frac{1}{N - N_{\text{trg}}} \sum_{l=N_{\text{trg}}+1}^{N} \frac{1}{T} \sum_{t=1}^{T} (\mathbb{E}_{\boldsymbol{X}}\left[ \boldsymbol{e}_k^\top \boldsymbol{W}_V^* \boldsymbol{x}_t \cdot (\boldsymbol{x}_t - \bar{\boldsymbol{x}}_{1:T}) | z_{T-1} = l \right]$$

$$- \mathbb{E}_{\boldsymbol{X}}\left[ \boldsymbol{e}_k^\top \boldsymbol{W}_V^* \boldsymbol{x}_t \cdot (\boldsymbol{x}_t - \bar{\boldsymbol{x}}_{1:T}) | z_{T-1} = l, z_{T+1} = k \right]) \boldsymbol{e}_l^\top.$$

Then the assertion immediately follows from the fact that each entry of

$$\mathbb{E}_{\boldsymbol{X}}\left[ \boldsymbol{e}_k^\top \boldsymbol{W}_V^* \boldsymbol{x}_t \cdot (\boldsymbol{x}_t - \bar{\boldsymbol{x}}_{1:T}) | z_{T-1} = l \right] - \mathbb{E}_{\boldsymbol{X}}\left[ \boldsymbol{e}_k^\top \boldsymbol{W}_V^* \boldsymbol{x}_t \cdot (\boldsymbol{x}_t - \bar{\boldsymbol{x}}_{1:T}) | z_{T-1} = l, z_{T+1} = k \right]$$

is upper bounded by $\tilde{\eta}_V \mathbb{E}[T^{-1}]$ up to constant, from Lemma 9. $\qquad \square$

### B.3 Max-sum Ratio and Algorithm Selection

Now we are ready to establish analysis on transformer's algorithm selection based on max-sum ratio. From Lemmas 11 and 12, it holds that

$$\boldsymbol{W}_{KQ}^*$$
$$= \tilde{\eta} \frac{1}{N_{\text{trg}}} \sum_{w=1}^{N_{\text{trg}}} \mathbb{E}\left[ \left\{ \begin{bmatrix} (T^{-1} + 2T^{-2})(\boldsymbol{p}_{\ell+2} + \boldsymbol{p}_{\ell+3}) \\ \mathbf{0} \\ T^{-1}\boldsymbol{e}_w \end{bmatrix} - 2T^{-2} \begin{bmatrix} \mathbf{1}_{1:T} \\ 2\boldsymbol{e}_w \\ \mathbf{0} \end{bmatrix} \right\} \begin{bmatrix} \boldsymbol{p}_T^\top & \boldsymbol{e}_w^\top & \mathbf{0}_N^\top \end{bmatrix} \right]$$
$$+ O_\infty(\tilde{\eta} N_{\text{trg}} \cdot N^{-1})$$

where $\tilde{\eta} = \tilde{\eta}_V \eta_{KQ} \mathbb{E}[T^{-1}]$.

Assume that a test sequence $z = [z_1, \ldots, z_{T^*}, z_{T^*+1}]$ is made from subtext lengths $(\ell_1^*, \ell_2^*)$ (hence $T^* = \ell_1^* + \ell_2^* + 3$). Now let $q_\lambda = \text{P}[\ell = \lambda]$ and $q^* = \text{P}[2\ell + 3 = T^*]$ respectively (probability is defined by pretraining distribution). If the trigger $\boldsymbol{x}_{T^*}$ satisfies $\boldsymbol{x}_{T^*} = w^*$, it holds that

$$\boldsymbol{W}_{KQ}^* \boldsymbol{x}_{T^*} = \tilde{\eta} \frac{q^*}{N_{\text{trg}}} \sum_{w=1}^{N_{\text{trg}}} \left( \begin{bmatrix} ((T^*)^{-1} + 2(T^*)^{-2})(\boldsymbol{p}_{\ell^*+2} + \boldsymbol{p}_{\ell^*+3}) \\ \mathbf{0} \\ (T^*)^{-1}\boldsymbol{e}_w \end{bmatrix} - 2(T^*)^{-2} \begin{bmatrix} \mathbf{1}_{1:T^*} \\ 2\boldsymbol{e}_w \\ \mathbf{0} \end{bmatrix} \right)$$

$$+ \tilde{\eta}\frac{1}{N_{\text{trg}}}\mathbb{E}\left[\begin{bmatrix}(T^{-1}+2T^{-2})(\boldsymbol{p}_{\ell+2}+\boldsymbol{p}_{\ell+3})\\\boldsymbol{0}\\T^{-1}\boldsymbol{e}_{w^*}\end{bmatrix}-2T^{-2}\begin{bmatrix}\boldsymbol{1}_{1:T}\\2\boldsymbol{e}_{w^*}\\\boldsymbol{0}\end{bmatrix}\right]$$

$$+ O_\infty(\tilde{\eta}N_{\text{trg}}\cdot N^{-1}),$$

where $\ell^* = (T^*-3)/2$ — if such $\ell^*$ is not an integer, then we do not define $\ell^*$ (in such case $q^* = 0$ holds and we don't need to define such a quantity).

Hence, for any $t$ we can calculate the attention logit as

$$s_t := \boldsymbol{x}_t^\top \boldsymbol{W}_{KQ}^* \boldsymbol{x}_{T^*}$$

$$= \tilde{\eta}\frac{q^*}{N_{\text{trg}}}\sum_{w=1}^{N_{\text{trg}}}\Big(((T^*)^{-1}+2(T^*)^{-2})(\mathbb{I}(t=\ell^*+2)+\mathbb{I}(t=\ell^*+3))$$

$$+ (T^*)^{-1}(\mathbb{I}(z_{t-1}=w)) - 2(T^*)^{-2}(\mathbb{I}(t\le T^*)+2\mathbb{I}(z_t=w))\Big)$$

$$+ \tilde{\eta}\frac{1}{N_{\text{trg}}}\Big((T(t-2)^{-1}+2T(t-2)^{-2})q_{t-2}+(T(t-3)^{-1}+2T(t-3)^{-2})q_{t-3}$$

$$+ \mathbb{E}[T^{-1}]\mathbb{I}(z_{t-1}=w^*)-2\mathbb{E}[T^{-2}\mathbb{I}(t\le T)]-4\mathbb{E}[T^{-2}]\mathbb{I}(z_t=w^*)\Big)$$

$$+ O_\infty(\tilde{\eta}N_{\text{trg}}\cdot N^{-1}). \tag{B.3}$$

We begin with showing that if max-sum ratio is not sufficiently large, we can construct an OOD test sequence $z^*$ such that transformer mistakenly use the positional shortcut:

**Lemma 13.** *If it holds that*

$$\frac{\max_\ell q_\ell \ell^{-1}}{\sum_\ell q_\ell \ell^{-1}} \ge \epsilon(N_{\text{trg}})$$

*where $\epsilon(N_{\text{trg}}) = \Theta(N_{\text{trg}}^{-1})$, there exists an OOD test sequence such that the pretrained transformer via Algorithm 1 fails to generalize.*

**Proof.** Assume that

$$z^* = [\underbrace{u,u,\ldots,u}_{\ell_1^*},w^*,v,\underbrace{u,u,\ldots,u}_{\ell_2^*},w^*,v].$$

and $T^* = \ell_1^* + \ell_2^* + 3 = 2\ell^* + 3$ where $\ell^* = \arg\max_\ell q(\ell)\ell^{-1}$. Furthermore, we assume

$$\ell_1^* \notin \{\ell^*-1, \ell^*, \ell^*+1, \ell^*+2\}. \tag{B.4}$$

Since we have Assumption 2, there exists $\ell_1^* \ge 1$ such that (B.4) holds. We show the following sub-lemma:

**Lemma 14.** *There exists $\epsilon_1(N_{\text{trg}}) = \Theta(N_{\text{trg}}^{-1})$ such that if*

$$\frac{\max_\ell q_\ell \ell^{-1}}{\sum_\ell q_\ell \ell^{-1}} \ge \epsilon_1(N_{\text{trg}}),$$

*then*

$$s_{\ell_1^*+1}, s_{\ell_1^*+2}, s_{\ell_1^*+3}, s_{T^*} \le \frac{1}{2}s_{\ell^*+2}.$$

**Proof.** Here we show $2s_{\ell_1^*+2} \le s_{\ell^*+2}$ — other properties can be deduced in the same vain.

Note that, from (B.3),

$$s_{\ell_1^*+2} \le \tilde{\eta}\frac{q^*}{N_{\text{trg}}}\sum_{w=1}^{N_{\text{trg}}}[(T^*)^{-1}\mathbb{I}(w=w^*)]$$

$$+ \frac{\tilde{\eta}}{N_{\text{trg}}}[((2\ell_1^*+3)^{-1}+2(2\ell_1^*+3)^{-2})q_{\ell_1^*}+((2\ell_1^*+1)^{-1}+2(2\ell_1^*+1)^{-2})q_{\ell_1^*-1}$$

$$+ \mathbb{E}[T^{-1}]] + O_\infty(\tilde{\eta} N_{\text{trg}} \cdot N^{-1})$$

$$\leq \frac{\tilde{\eta} q^*}{N_{\text{trg}}} (T^*)^{-1} + \frac{6\tilde{\eta} q^*}{N_{\text{trg}}} (T^*)^{-1} + \frac{\tilde{\eta}}{N_{\text{trg}}} \mathbb{E}[T^{-1}] + O_\infty(\tilde{\eta} N_{\text{trg}} \cdot N^{-1}).$$

On the other hand, it holds that

$$s_{\ell^*+2}$$

$$\geq \tilde{\eta} \frac{q^*}{N_{\text{trg}}} \sum_{w=1}^{N_{\text{trg}}} [(T^*)^{-1} + 2(T^*)^{-2} - 2(T^*)^{-2}(\mathbb{I}(\ell^* + 2 \leq T^*) + 2\mathbb{I}(z_{\ell^*+2} = w))]$$

$$+ \frac{\tilde{\eta}}{N_{\text{trg}}} [((2\ell^* + 3)^{-1} + 2(2\ell^* + 3)^{-2}) q_{\ell^*} + ((2\ell^* + 1)^{-1} + 2(2\ell^* + 1)^{-2}) q_{\ell^*-1} - 6\mathbb{E}[T^{-2}]]$$

$$+ O_\infty(\tilde{\eta} N_{\text{trg}} \cdot N^{-1})$$

$$\geq \tilde{\eta} q^* (T^*)^{-1} - 6\tilde{\eta} q^* (T^*)^{-2} - 6\frac{\tilde{\eta}}{N_{\text{trg}}} \mathbb{E}[T^{-2}] + O_\infty(\tilde{\eta} N_{\text{trg}} \cdot N^{-1}).$$

Note that $T^{-2} \leq \frac{1}{10} T^{-1}$ holds from Assumption 2. Therefore,

$$\frac{1}{2} s_{\ell^*+2} - s_{\ell_1^*+2} \geq \frac{1}{5} \tilde{\eta} q^* (T^*)^{-1} - \frac{13}{10} \frac{\tilde{\eta}}{N_{\text{trg}}} \mathbb{E}[T^{-1}] - \frac{7\tilde{\eta} q^*}{N_{\text{trg}}} (T^*)^{-1} + O_\infty(\tilde{\eta} N_{\text{trg}} \cdot N^{-1}).$$

Together with Assumption 1, if the max-sum ratio is $\Omega(N_{\text{trg}}^{-1})$, we obtain $2s_{\ell_1^*+2} \leq s_{\ell^*+2}$ as desired. □

Since now we have Lemma 14, when

$$\tilde{\eta}_V \eta_{KQ} \gtrsim C \log N \frac{N^2}{N_{\text{trg}}^3} \gtrsim C \log N \cdot \frac{N_{\text{trg}}}{\mathbb{E}[T^{-1}]^2} \tag{B.5}$$

for a sufficiently large $C$ we obtain $\exp s_t / \exp s_{\ell^*+2} \leq \exp\{-C \log N\} = N^{-C}$ where $t = \ell_1^* + 1, \ell_1^* + 2, \ell_1^* + 3$ and $T^*$. This immediately implies that $\boldsymbol{X}_{1:T^*} \text{Softmax}(\boldsymbol{X}_{1:T^*}^\top \boldsymbol{W}_{KQ} \boldsymbol{x}_{T^*}) = \sum_{k \neq \ell^*+1, \ell^*+2, \ell^*+3, T^*} \alpha_k \boldsymbol{x}_k + O_\infty(N^{-C'})$ for a sufficiently large $C'$ where $\sum_{k \neq \ell^*+1, \ell^*+2, \ell^*+3, T^*} \alpha_k \geq 1 - N^{-C'}$. Therefore, we get

$$\boldsymbol{X}_{1:T^*} \text{Softmax}(\boldsymbol{X}_{1:T^*}^\top \boldsymbol{W}_{KQ} \boldsymbol{x}_{T^*}) = (1 - N^{-C'}) \begin{bmatrix} * \\ \boldsymbol{e}_u \\ \boldsymbol{e}_u \end{bmatrix} + N^{-C'} \begin{bmatrix} * \\ * \\ * \end{bmatrix}.$$

From the structure of $\boldsymbol{W}_V^*$ (Lemma 9), we observe that the predicted logit $\boldsymbol{W}_V \boldsymbol{X}_{1:T^*} \text{Softmax}(\boldsymbol{X}_{1:T^*}^\top \boldsymbol{W}_{KQ} \boldsymbol{x}_{T^*})$ has a peak on the token $u$, meaning that OOD generalization fails. □

**Remark 3.** *Here we worked on the ratio between* $q_{\ell^*} T(\ell^*)^{-1} = \max_\ell (2\ell + 3)^{-1} q_\ell$ *and* $\mathbb{E}[T(\ell)^{-1}] = \sum_\ell (2\ell + 3)^{-1} q_\ell$. *We can immediately show (using Assumption 2)* $\frac{1}{3} \max_\ell (\ell)^{-1} q_\ell \leq \max_\ell (2\ell + 3)^{-1} q_\ell \leq \frac{1}{2} \max_\ell (\ell)^{-1} q_\ell$ *and* $\frac{1}{3} \sum_\ell (\ell)^{-1} q_\ell \leq \sum_\ell (2\ell + 3)^{-1} q_\ell \leq \frac{1}{2} \sum_\ell (\ell)^{-1} q_\ell$, *then we do not distinguish these two definitions of max-sum ratio.*

Similarly we can show the following upper bound:

**Lemma 15.** *If it holds that*

$$\frac{\max_\ell q_\ell \ell^{-1}}{\sum_\ell q_\ell \ell^{-1}} \geq \epsilon(N_{\text{trg}})$$

*where* $\epsilon(N_{\text{trg}}) = \Theta(N_{\text{trg}}^{-1})$, *the pretrained transformer via Algorithm 1 can generalize OOD.*

**Proof.** In the same vain as the proof of the lower bound, it suffices to show $s_{\ell_1^*+2} \geq 2s_t$ for any $t \neq \ell_1^* + 2$, going the other way around Lemma 13.

First we have

$$s_{\ell_1^*+2} \geq \tilde{\eta} \frac{q^*}{N_{\text{trg}}} \sum_{w=1}^{N_{\text{trg}}} [-2(T^*)^{-2} \mathbb{I}(\ell_1^* + 2 \leq T^*)]$$

$$+ \frac{\tilde{\eta}}{N_{\text{trg}}}[((2\ell_1^* + 3)^{-1} + 2(2\ell_1^* + 3)^{-2})q_{\ell_1^*} + ((2\ell_1^* + 1)^{-1} + 2(2\ell_1^* + 1)^{-2})q_{\ell_1^* - 1}$$

$$- 2\mathbb{E}[T^{-2}] + \mathbb{E}[T^{-1}]] + O_\infty(\tilde{\eta}N_{\text{trg}} \cdot N^{-1})$$

$$\geq \frac{\tilde{\eta}}{N_{\text{trg}}}\mathbb{E}[T^{-1}] - 2\frac{\tilde{\eta}}{N_{\text{trg}}}\mathbb{E}[T^{-2}] - 2\tilde{\eta}\max_\ell q_\ell T(\ell)^{-1} + O_\infty(\tilde{\eta}N_{\text{trg}} \cdot N^{-1}).$$

For all $t \neq \ell_1^* + 2$, observe

$$s_t \leq \tilde{\eta}\frac{q^*}{N_{\text{trg}}}\sum_{w=1}^{N_{\text{trg}}}[3(T^*)^{-1} \cdot 2] + \frac{\tilde{\eta}}{N_{\text{trg}}}[3T(t-2)^{-1}q_{t-2} + 3T(t-3)^{-1}q_{t-3}] + O_\infty(\tilde{\eta}N_{\text{trg}} \cdot N^{-1})$$

$$\leq 6\tilde{\eta}\max_\ell q_\ell T(\ell)^{-1} + 6\frac{\tilde{\eta}}{N_{\text{trg}}}\max_\ell q_\ell T(\ell)^{-1} + O_\infty(\tilde{\eta}N_{\text{trg}} \cdot N^{-1}).$$

Therefore, we obtain $\frac{1}{2}s_{\ell_1^* + 2} \geq s_t$ as desired, if max-sum ratio is $O(N_{\text{trg}}^{-1})$. $\qquad\square$

## B.4  Proof of Theorem 5

Theorem 5 is directly obtained by combining Lemmas 13 and 15: it only remains to adjust the stepsize.

From Corrorally 10 we need $\tilde{\eta}_V = \eta_V/N \lesssim 1/N$, and from (B.5) we need $\tilde{\eta}_V\eta_{KQ} \gtrsim \log N \frac{N^2}{N_{\text{trg}}^3}$. Therefore, it suffices to set

$$\eta_V \lesssim 1 \text{ and } \eta_V\eta_{KQ} \gtrsim \frac{N^3}{N_{\text{trg}}^3}\log N.$$

# C  Finite Sample Analysis

Now we turn to make an analysis for finite sample size setting.

**Proof Sketch.** We explain how to evaluate the concentration of the empirical gradient $\nabla_{\boldsymbol{W}_V}\hat{\mathcal{L}}(f_{\text{TF}})$. Concentration for $\nabla_{\boldsymbol{W}_{KQ}}\hat{\mathcal{L}}(f_{\text{TF}})$ can be obtained similarly.

Let $\left\{\left\{\boldsymbol{x}_t^{(i)}\right\}_{t=1}^{T(\ell)_i}\right\}_{i=1}^{M_V}$ be $M_V$ i.i.d. sample sequences, and $\left\{\left\{\boldsymbol{x}_t^{(i_{\ell,k})}\right\}_t\right\}_{i_{\ell,k}=1}^{M_V^{\ell,k}}$ be $M_V^{\ell,k}$ i.i.d. sub-samples conditioned on $\ell$ and $z_{T+1} = k$. Note that $\sum_{\ell,k} M_V^{\ell,k} = M_V$. The empirical gradient $\nabla_{\boldsymbol{W}_V}\hat{\mathcal{L}}(f_{\text{TF}})$ is expressed as

$$\nabla_{\boldsymbol{W}_V}\hat{\mathcal{L}}(f_{\text{TF}}) \simeq \hat{\mathbb{E}}_{\ell \sim \mathcal{D}_\ell}\hat{\mathbb{E}}_{k \sim \text{Unif}[K]}\left[\hat{\boldsymbol{A}}_{\ell,k}\right] + \text{(similar terms omitted)}$$

where $\hat{\boldsymbol{A}}_{\ell,k}$ is the empirical average of $\frac{1}{T}\sum_{t=1}^T \mathbf{1}_N \boldsymbol{x}_t^{(i_{\ell,k})\top}$ ($i_{\ell,k} = 1, \ldots, M_V^{\ell,k}$) with the sample size $= M_V^{\ell,k}$. We focus on bounding the first term in this sketch. The gap between the empirical and population gradients is bounded as

$$\|\nabla_{\boldsymbol{W}_V}\hat{\mathcal{L}}(f_{\text{TF}}) - \nabla_{\boldsymbol{W}_V}\mathcal{L}(f_{\text{TF}})\|_2$$
$$\lesssim \left\|\hat{\mathbb{E}}_\ell\hat{\mathbb{E}}_k\left[\hat{\boldsymbol{A}}_{\ell,k} - \mathbb{E}[\hat{\boldsymbol{A}}_{\ell,k}|\ell,k]\right]\right\|_2 + \left\|\hat{\mathbb{E}}_\ell\left(\hat{\mathbb{E}}_k - \mathbb{E}_k\right)\mathbb{E}[\hat{\boldsymbol{A}}_{\ell,k}|\ell,k]\right\|_2 + \left\|\left(\hat{\mathbb{E}}_\ell - \mathbb{E}_\ell\right)\mathbb{E}[\hat{\boldsymbol{A}}_{\ell,k}|\ell]\right\|_2.$$

The first term is bounded by the matrix Hoeffding's inequality using $\|\hat{\boldsymbol{A}}_{\ell,k}\hat{\boldsymbol{A}}_{\ell,k}^\top\|_2, \|\hat{\boldsymbol{A}}_{\ell,k}^\top\hat{\boldsymbol{A}}_{\ell,k}\|_2 \lesssim 1$ and $M_V^{\ell,k} \simeq q_\ell N^{-1}M_V$ for each pair $(\ell, k)$. Note that $\sum_\ell \sqrt{q_\ell}$ emerges in this bound because $\|\hat{\boldsymbol{A}}_{\ell,k} - \mathbb{E}[\hat{\boldsymbol{A}}_{\ell,k}|\ell,k]\|_2 \simeq q_\ell^{-1/2}$ and $\hat{\mathbb{E}}_\ell[\cdot] \simeq \sum_\ell \cdot q_\ell$. The second and third terms can also be bounded by $\|\hat{\boldsymbol{A}}_{\ell,k}\|_2 \lesssim 1$ and using the standard Hoeffding's inequality.

## C.1 Value Matrix

We first establish an upper-bound for the difference between empirical and population gradient with respect to $\boldsymbol{W}_V$.

**Lemma 16.** *Let* $\boldsymbol{A}_t = \boldsymbol{1}_{1:N}\boldsymbol{x}_t^\top = \boldsymbol{1}_{1:N}[\boldsymbol{p}_t^\top \ \boldsymbol{e}_{z_t}^\top \ \boldsymbol{e}_{z_{t-1}}^\top]$. *Then we have*

$$\begin{bmatrix} \boldsymbol{A}_t\boldsymbol{A}_t^\top & \boldsymbol{0} \\ \boldsymbol{0} & \boldsymbol{A}_t^\top\boldsymbol{A}_t \end{bmatrix} \preceq \boldsymbol{\Sigma}_t$$

*where* $\lambda_i(\boldsymbol{\Sigma}_t) \lesssim N$ *for* $i = 1, 2$ $\lambda_i(\boldsymbol{\Sigma}_t) = 0$ *for* $i \geq 3$ *and* $\|\boldsymbol{\Sigma}_t\|_2 \lesssim N$. *Similarly, let* $\boldsymbol{B}_{t,k} = \boldsymbol{e}_k\boldsymbol{x}_t^\top$. *Then, we have*

$$\begin{bmatrix} \boldsymbol{B}_{t,k}\boldsymbol{B}_{t,k}^\top & \boldsymbol{0} \\ \boldsymbol{0} & \boldsymbol{B}_{t,k}^\top\boldsymbol{B}_{t,k} \end{bmatrix} \preceq \boldsymbol{\Sigma}_t'$$

*where* $\|\boldsymbol{\Sigma}_t'\|_2 \lesssim 1$.

**Proof.** We provide the proof for $\boldsymbol{A}_t$. Proof for $\boldsymbol{B}_t$ can be derived in the same vain.

$$\boldsymbol{A}_t\boldsymbol{A}_t^\top = \boldsymbol{1}_{1:N}\boldsymbol{x}_t^\top\boldsymbol{x}_t\boldsymbol{1}_{1:N}^\top = 3\begin{bmatrix} 1 & \cdots & 1 \\ \vdots & \ddots & \vdots \\ 1 & \cdots & 1 \end{bmatrix} \preceq \boldsymbol{\Sigma}_t^{(a)}$$

where $\lambda_1(\boldsymbol{\Sigma}_t^{(a)}) = 3N$, $\lambda_i(\boldsymbol{\Sigma}_t^{(a)}) = 0$ for $i \geq 2$.

$$\boldsymbol{A}_t^\top\boldsymbol{A}_t = \boldsymbol{x}_t\boldsymbol{1}_{1:N}^\top\boldsymbol{1}_{1:N}\boldsymbol{x}_t^\top = N\boldsymbol{x}_t\boldsymbol{x}_t^\top \preceq \boldsymbol{\Sigma}_t^{(b)}$$

where $\lambda_1(\boldsymbol{\Sigma}_t^{(b)}) = 3N$, $\lambda_i(\boldsymbol{\Sigma}_t^{(b)}) = 0$ for $i \geq 2$. Using

$$\operatorname{rank}(\boldsymbol{A}_t\boldsymbol{A}_t^\top) + \operatorname{rank}(\boldsymbol{A}_t^\top\boldsymbol{A}_t) \leq 2$$

and

$$\lambda_1\left(\begin{bmatrix} \boldsymbol{A}_t\boldsymbol{A}_t^\top & \boldsymbol{O} \\ \boldsymbol{O} & \boldsymbol{A}_t^\top\boldsymbol{A}_t \end{bmatrix}\right) \lesssim \max\{\lambda_1(\boldsymbol{A}_t\boldsymbol{A}_t^\top), \lambda_1(\boldsymbol{A}_t^\top\boldsymbol{A}_t)\},$$

we obtain the conclusion. $\qquad\square$

**Lemma 17.** *Let* $\left\{\left\{\boldsymbol{x}_t^{(i)}\right\}_{t=1}^{T_i}\right\}_{i=1}^{M_V}$ *be i.i.d. sample sequences,* $\left\{\left\{\boldsymbol{x}_t^{(i_T)}\right\}_{t=1}^{T}\right\}_{i_T=1}^{M_V^T}$ *be conditionally i.i.d. sub-samples with fixed* $T$, *and* $\left\{\left\{\boldsymbol{x}_t^{(i_{T,k})}\right\}_t\right\}_{i_{T,k}=1}^{M_V^{T,k}}$ *be conditionally i.i.d. sub-samples with fixed* $T$ *and* $z_{T+1} = k$. *Note that* $\sum_{T,k} M_V^{T,k} = \sum_T M_V^T = M_V$. *Then, with probability at least* $1 - O(\epsilon)$,

$$\lambda_{\max}\left(\nabla_{\boldsymbol{W}_V}\frac{1}{M_V}\sum_{i=1}^{M_V}\mathcal{L}(\boldsymbol{X}_{1:T^{(i)}}^{(i)}; \boldsymbol{W}_{KQ}, \boldsymbol{W}_V) - \nabla_{\boldsymbol{W}_V}\bar{\mathcal{L}}(\boldsymbol{W}_{KQ}, \boldsymbol{W}_V)\right)$$

$$\lesssim \left(\sum_\ell \sqrt{q_\ell}\right)\sqrt{\frac{N\log\left(NL(N+L)\epsilon^{-1}\right)}{M_V}}.$$

**Proof.** The empirical gradient is, noting that $T = 2\ell + 3 \geq 5$,

$$\nabla_{\boldsymbol{W}_V}\hat{\mathcal{L}}(f_{\mathrm{TF}}) = \frac{1}{N}\sum_{k=1}^{N}\boldsymbol{e}_k\hat{\mathbb{E}}_T\left[\frac{1}{T}\sum_{t=1}^{T}\hat{\mathbb{E}}[\boldsymbol{x}_t]^\top\right]$$

$$- \sum_{k=N_{\mathrm{trg}}+1}^{N}\boldsymbol{e}_k\hat{\mathbb{E}}_T\left[\frac{1}{T}\sum_{t=1}^{T}\hat{\mathbb{E}}[\boldsymbol{x}_t\mathbb{I}[z_{T+1}=k]]^\top\right]$$

$$= \frac{1}{M_V}\sum_{T=5}^{L-1}\sum_{i_T=1}^{M_V^T}\left(\frac{1}{TN}\sum_{t=1}^{T}\boldsymbol{1}_{1:N}\boldsymbol{x}_t^{(i_T)\top}\right)$$

$$-\sum_{T=5}^{L-1}\frac{M_V^T}{M_V}\sum_{k=N_{\text{trg}}+1}^{N}\frac{M_V^{T,k}}{M_V^T}\frac{1}{M_V^{T,k}}\sum_{i_{T,k}=1}^{M_V^{T,k}}\left(\frac{1}{T}\sum_{t=1}^{T}\boldsymbol{e}_k\boldsymbol{x}_t^{(i_{T,k})^\top}\right).$$

Let

$$\boldsymbol{A}^{(i_T)}=\sum_{t=1}^{T}\mathbf{1}_{1:N}\boldsymbol{x}_t^{(i_T)^\top}-\mathbb{E}\left[\sum_{t=1}^{T}\mathbf{1}_{1:N}\boldsymbol{x}_t^{(i_T)^\top}\mid T\right]$$

and

$$\boldsymbol{B}^{(i_{T,k})}=\sum_{t=1}^{T}\boldsymbol{e}_k\boldsymbol{x}_t^{(i_{T,k})^\top}-\mathbb{E}\left[\sum_{t=1}^{T}\boldsymbol{e}_k\boldsymbol{x}_{t}^{(i_{T,k})^\top}\mid y=k,T\right].$$

Using Lemma 16, we can bound the operator norms as

$$\frac{1}{T^2}\left\|\begin{bmatrix}\boldsymbol{A}^{(i_T)}\boldsymbol{A}^{(i_T)^\top}&\boldsymbol{O}\\\boldsymbol{O}&\boldsymbol{A}^{(i_T)^\top}\boldsymbol{A}^{(i_T)}\end{bmatrix}\right\|_2\lesssim\sqrt{N}$$

and

$$\frac{1}{T^2}\left\|\begin{bmatrix}\boldsymbol{B}^{(i_{T,k})}\boldsymbol{B}^{(i_{T,k})^\top}&\boldsymbol{O}\\\boldsymbol{O}&\boldsymbol{B}^{(i_{T,k})^\top}\boldsymbol{B}^{(i_{T,k})}\end{bmatrix}\right\|_2\lesssim1.$$

By combining the matrix Hoeffding's inequality and the union bound, we have

$$\left\|\frac{1}{M_V^T T N}\sum_{i_T=1}^{M_V^T}\boldsymbol{A}^{(i_T)}\right\|_2\lesssim\frac{1}{\sqrt{NM_V^T}}\sqrt{\log\left(L(N+L)\epsilon^{-1}\right)}$$

and

$$\left\|\frac{1}{M_V^{T,k}}\sum_{i_{T,k}=1}^{M_V^{T,k}}\frac{1}{T^{(i_{T,k})}}\boldsymbol{B}^{(i_{T,k})}\right\|_2\lesssim\frac{1}{\sqrt{M_V^{T,k}}}\sqrt{\log\left(NL(N+L)\epsilon^{-1}\right)}.$$

with probability at least $1-O(\epsilon)$. Using the union bound $M_V^T=M_V q_{T(\ell)}(1\pm M_V^{-1/2}q_{T(\ell)}^{-1/2}\sqrt{\log(L\epsilon^{-1})})$, the matrix Hoeffding's inequality, standard Hoeffding's inequality, and $\|\mathbf{1}_{1:N}\boldsymbol{x}_t^{(i_T)^\top}\|_2\lesssim1$, we obtain

$$\left\|\frac{1}{M_V}\sum_{T=5}^{L-1}\sum_{i_T=1}^{M_V^T}\left(\frac{1}{TN}\sum_{t=1}^{T}\mathbf{1}_{1:N}\boldsymbol{x}_t^{(i_T)^\top}\right)-\mathbb{E}\left(\frac{1}{TN}\sum_{t=1}^{T}\mathbf{1}_{1:N}\boldsymbol{x}_t^{(i_T)^\top}\right)\right\|_2$$

$$\lesssim\left\|\sum_{T=5}^{L-1}\frac{M_V^T}{M_V}\frac{1}{M_V^T}\sum_{i_T=1}^{M_V^T}\frac{1}{TN}\boldsymbol{A}^{(i_T)}\right\|_2+\left\|\sum_{T=5}^{L-1}\left(\frac{M_V^T}{M_V}-q_{T(\ell)}\right)\mathbb{E}\left(\frac{1}{TN}\sum_{t=1}^{T}\mathbf{1}_{1:N}\boldsymbol{x}_t^{(i_T)^\top}\mid T\right)\right\|_2$$

$$\lesssim\sum_{T=5}^{L-1}\underbrace{\frac{M_V^T}{M_V}}_{\simeq\text{Prob}(T)}\underbrace{\left\|\frac{1}{M_V^T}\sum_{i_T=1}^{M_V^T}\frac{1}{TN}\boldsymbol{A}^{(i_T)}\right\|_2}_{\sqrt{\frac{\log\left(L(N+L)\epsilon^{-1}\right)}{NM_V\text{Prob}(T)}}}+\sum_{T=5}^{L-1}\underbrace{\left|\frac{M_V^T}{M_V}-q_{T(\ell)}\right|}_{\lesssim\frac{\sqrt{q_{T(\ell)}\log(L\epsilon^{-1})}}{\sqrt{M_V}}}\underbrace{\left\|\mathbb{E}\left(\frac{1}{TN}\sum_{t=1}^{T}\mathbf{1}_{1:N}\boldsymbol{x}_t^{(i_T)^\top}\mid T\right)\right\|_2}_{\lesssim N^{-1/2}}$$

$$\lesssim\frac{\sum_\ell\sqrt{q_\ell}}{\sqrt{NM_V}}\sqrt{\log\left(L(N+L)\epsilon^{-1}\right)}.$$

Let $\bar{\boldsymbol{B}}^{(i_{T,k})}=\sum_{t=1}^{T}\boldsymbol{e}_k\boldsymbol{x}_t^{(i_{T,k})^\top}$. Using $M_V^{T,k}\simeq M_V^T(N-N_{\text{trg}})^{-1}(1\pm(M_V^T)^{-1/2}N^{1/2}\sqrt{\log(NL\epsilon^{-1})})$, $M_V^T=M_V q_{T(\ell)}(1\pm M_V^{-1/2}q_{T(\ell)}^{-1/2}\sqrt{\log(L\epsilon^{-1})})$, the matrix Hoeffding's inequality, standard Hoeffding's inequality, and $\|T^{-1}\bar{\boldsymbol{B}}^{(i_{T,k})}\|_2\lesssim1$, we obtain

$$\left\|\sum_{T=5}^{L-1}\frac{M_V^T}{M_V}\sum_{k=N_{\text{trg}}+1}^{N}\frac{M_V^{T,k}}{M_V^T}\frac{1}{M_V^{T,k}}\sum_{i_{T,k}=1}^{M_V^{T,k}}\left(\frac{1}{T}\sum_{t=1}^{T}\boldsymbol{e}_k\boldsymbol{x}_t^{(i_{T,k})^\top}\right)\right.$$

$$
-\sum_{k=N_{\text{trg}}+1}^{N} \frac{1}{N-N_{\text{trg}}} \mathbb{E}_T \left[ \mathbb{E}\left[ \left( \frac{1}{T} \sum_{t=1}^{T} \boldsymbol{e}_k \boldsymbol{x}_t^{(i_{T,k})\top} \right) \mid y=k \right] \right] \Bigg\|_2
$$

$$
\lesssim \left\| \sum_{T=5}^{L-1} \frac{M_V^T}{M_V} \sum_{k=N_{\text{trg}}+1}^{N} \frac{M_V^{T,k}}{M_V^T} \frac{1}{M_V^{T,k}} \sum_{i_{T,k}=1}^{M_V^{T,k}} T^{-1} \boldsymbol{B}^{(i_{T,k})} \right\|_2
$$

$$
+ \left\| \sum_{T=5}^{L-1} \frac{M_V^T}{M_V} \sum_{k=N_{\text{trg}}+1}^{N} \left( \frac{M_V^{T,k}}{M_V^T} - \frac{1}{N-N_{\text{trg}}} \right) \frac{1}{T} \mathbb{E}\left[ \bar{\boldsymbol{B}}^{(i_{T,k})} \mid y=k, T \right] ) \right\|_2
$$

$$
+ \left\| \sum_{T=5}^{L-1} \left( \frac{M_V^T}{M_V} - q_{T(\ell)} \right) \sum_{k=N_{\text{trg}}+1}^{N} \frac{1}{N-N_{\text{trg}}} \mathbb{E}[T^{-1}\bar{\boldsymbol{B}}^{(i_{T,k})}|k,T] \right\|_2
$$

$$
\lesssim \sum_{T=5}^{L-1} \underbrace{\frac{M_V^T}{M_V}}_{q_{T(\ell)}} \sum_{k=N_{\text{trg}}+1}^{N} \underbrace{\frac{M_V^{T,k}}{M_V^T}}_{N^{-1}} \underbrace{\left\| \frac{1}{M_V^{T,k}} \sum_{i_{T,k}=1}^{M_V^{T,k}} T^{-1} \boldsymbol{B}^{(i_{T,k})} \right\|_2}_{\lesssim \sqrt{\frac{N\log\left(NL(N+L)\epsilon^{-1}\right)}{M_V\,\text{Prob}(T)}}}
$$

$$
+ \sum_{T=5}^{L-1} \frac{M_V^T}{M_V} \sum_{k=N_{\text{trg}}+1}^{N} \underbrace{\left| \frac{M_V^{T,k}}{M_V^T} - \frac{1}{N-N_{\text{trg}}} \right|}_{\lesssim \sqrt{\frac{\log\left(NL\epsilon^{-1}\right)}{NM_V\,q_{T(\ell)}}}} \underbrace{\left\| \frac{1}{T} \mathbb{E}\left[ \bar{\boldsymbol{B}}^{(i_{T,k})} \mid y=k, T \right] ) \right\|_2}_{\lesssim 1}
$$

$$
+ \sum_{T=5}^{L-1} \left| \frac{M_V^T}{M_V} - q_{T(\ell)} \right| \left\| \sum_{k=N_{\text{trg}}+1}^{N} \frac{1}{N-N_{\text{trg}}} \mathbb{E}[T^{-1}\bar{\boldsymbol{B}}^{(i_{T,k})}|k,T] \right\|_2
$$

$$
\lesssim \left( \sum_\ell \sqrt{q_\ell} \right) \sqrt{ \frac{N\log\left(NL(N+L)\epsilon^{-1}\right)}{M_V} }.
$$

$\square$

Note that if $M_V \gtrsim \text{poly}\log N \cdot \frac{N^5}{N_{\text{trg}}^2}\left( \frac{\sum_{\ell\in\mathcal{S}}\sqrt{q_\ell}}{\sum_{\ell\in\mathcal{S}} q_\ell \ell^{-1}} \right)^2$, then we obtain $\|\bar{\boldsymbol{W}}_V^* - \boldsymbol{W}_V^*\|_2 \lesssim \tilde{\eta}_V \mathbb{E}[T^{-1}]N_{\text{trg}}N^{-1}$, where $\bar{\boldsymbol{W}}_V^*$ and $\boldsymbol{W}_V^*$ are $\boldsymbol{W}_V$ after one GD step with infinite and finite sample size, respectively. The following corollary is obtained by combining Lemmas 9 and 17.

**Corollary 18.** *If we set $\eta_V \lesssim 1$ and $M_V \gtrsim \text{poly}\log N \cdot \frac{N^5}{N_{\text{trg}}^2}\left( \frac{\sum_{\ell\in\mathcal{S}}\sqrt{q_\ell}}{\sum_{\ell\in\mathcal{S}} q_\ell \ell^{-1}} \right)^2$, then it holds that $|1/N - p(k|z)| = O(\mathbb{E}[T^{-1}]/N^2)$ for any $k$ and $z$, where $p(k|z)$ is the transformer output regarding token $k$ after pretraining $\boldsymbol{W}_V$.*

### C.2 Key-Query matrix

Now let us consider the KQ-matrix with finite samples

$$
\boldsymbol{W}_{KQ} = -\eta_{KQ} \left( \sum_{T=5}^{L-1} \frac{M_{KQ}^T}{M_{KQ}} \frac{1}{M_{KQ}^T} \sum_{i_T} \hat{\boldsymbol{C}}_{i_T} - \sum_{T=5}^{L-1} \sum_{k=N_{\text{trg}}+1}^{N} \frac{M_{KQ}^T}{M_{KQ}} \frac{M_{KQ}^{T,k}}{M_{KQ}^T} \frac{1}{M_{KQ}^{T,k}} \sum_{i_{T,k}} \hat{\boldsymbol{D}}_{i_{T,k}} \right)
$$
$$
+ \eta_{KQ}\tilde{\eta}_V \mathbb{E}[T^{-1}] O_\infty(N_{\text{trg}}/N)
$$

where

$$
\hat{\boldsymbol{C}}_{i_T} := \sum_{k=N_{\text{trg}}+1}^{N} \frac{1}{N-N_{\text{trg}}} \left( \left( \frac{1}{T} - \frac{1}{T^2} \right) \sum_t \boldsymbol{x}_t^{(i_T)} \boldsymbol{x}_t^{(i_T)\top} \boldsymbol{W}_V^\top \boldsymbol{e}_k \boldsymbol{x}_T^{(i_T)\top} \right)
$$
$$
- \sum_{k=N_{\text{trg}}+1}^{N} \frac{1}{N-N_{\text{trg}}} \left( \frac{1}{T^2} \sum_{t\neq t'} \boldsymbol{x}_t^{(i_T)} \boldsymbol{x}_{t'}^{(i_T)\top} \boldsymbol{W}_V^\top \boldsymbol{e}_k \boldsymbol{x}_T^{(i_T)\top} \right)
$$

and

$$\hat{\boldsymbol{D}}_{i_{T,k}} := \left(\left(\frac{1}{T} - \frac{1}{T^2}\right)\sum_t \boldsymbol{x}_t^{(i_{T,k})}\boldsymbol{x}_t^{(i_{T,k})\top}\boldsymbol{W}_V^\top\boldsymbol{e}_k\boldsymbol{x}_T^{(i_{T,k})\top}\right)$$
$$- \left(\frac{1}{T^2}\sum_{t\neq t'}\boldsymbol{x}_t^{(i_{T,k})}\boldsymbol{x}_{t'}^{(i_{T,k})\top}\boldsymbol{W}_V^\top\boldsymbol{e}_k\boldsymbol{x}_T^{(i_{T,k})\top}\right)$$

where

$$\boldsymbol{W}_V^\top\boldsymbol{e}_k = \tilde{\eta}_V\mathbb{E}[T^{-1}]\begin{bmatrix}0\\ \vdots\\ 0\\ \hline \boldsymbol{0}_{k-1}\\ 1\\ \boldsymbol{0}_{N-k}\\ \hline \boldsymbol{0}_{k-1}\\ 1\\ \boldsymbol{0}_{N-k}\end{bmatrix} + o(\tilde{\eta}_V\mathbb{E}[T^{-1}])\cdot\boldsymbol{1}_{L+2N}.$$

**Lemma 19.** *Let $\boldsymbol{B}_{t,t',T,k} = \boldsymbol{x}_t(\boldsymbol{x}_{t'}^\top\boldsymbol{W}_V\boldsymbol{e}_k)\boldsymbol{x}_T^\top$. Then, we have*

$$\boldsymbol{B}_{t,t',T,k} = \left((\mathbb{1}[z_{t'} = k] + \mathbb{1}[z_{t'-1} = k])\tilde{\eta}_V\mathbb{E}[T^{-1}] + o(\tilde{\eta}_V\mathbb{E}[T^{-1}])\right)\boldsymbol{x}_t\boldsymbol{x}_T^\top.$$

*Therefore,*

$$\begin{bmatrix}\boldsymbol{B}_{t,t',T,k}\boldsymbol{B}_{t,t',T,k}^\top & \boldsymbol{O}\\ \boldsymbol{O} & \boldsymbol{B}_{t,t',k}^\top\boldsymbol{B}_{t,t',k}\end{bmatrix} \preceq \boldsymbol{\Sigma}$$

*where $\lambda_i(\boldsymbol{\Sigma}) = O(\tilde{\eta}_V\mathbb{E}[T^{-1}])^2$ for $i = 1, 2$ and $\lambda_i(\boldsymbol{\Sigma}) = 0$ for $i > 3$.*

**Proof.** The proof is straightforward. Note that $\boldsymbol{x}_{t'}^\top\boldsymbol{1}_{L+2N} = O(1)$ for all $t'$. $\qquad\square$

**Lemma 20.** *Let $\bar{\boldsymbol{W}}_{KQ}^*$ be $\boldsymbol{W}_{KQ}$ after one gradient descent step with finite sample size (Algorithm 1) and $\boldsymbol{W}_{KQ}^*$ be the counterpart for infinite sample size. Let $\left\{\left\{\boldsymbol{x}_t^{(i)}\right\}_{t=1}^{T_i}\right\}_{i=1}^{M_V}$ be i.i.d. sample sequences, $\left\{\left\{\boldsymbol{x}_t^{(i_T)}\right\}_{t=1}^{T}\right\}_{i_t=1}^{M_V^T}$ be conditionally i.i.d. sub-samples with fixed $T$, and $\left\{\left\{\boldsymbol{x}_t^{(i_{T,k})}\right\}_t\right\}_{i_{T,k}=1}^{M_V^{T,k}}$ be conditionally i.i.d. sub-samples with fixed $T$ and $z_{T+1} = k$. With probability at least $1 - O(\epsilon)$,*

$$\lambda_{\max}(\bar{\boldsymbol{W}}_{KQ}^* - \boldsymbol{W}_{KQ}^*) \lesssim \eta_{KQ}\tilde{\eta}_V\mathbb{E}[T^{-1}]\left(\sum_l\sqrt{q_l}\right)\sqrt{\frac{N\log\left(LN(L+N)\epsilon^{-1}\right)}{M_{KQ}}}$$
$$+ \eta_{KQ}\tilde{\eta}_V\mathbb{E}[T^{-1}]O_\infty(N_{\text{trg}}/N).$$

**Proof.** We have

$$\left\|\begin{bmatrix}\hat{\boldsymbol{C}}_{i_T}\hat{\boldsymbol{C}}_{i_T}^\top & \boldsymbol{O}\\ \boldsymbol{O} & \hat{\boldsymbol{C}}_{i_T}^\top\hat{\boldsymbol{C}}_{i_T}\end{bmatrix}\right\|_2 \lesssim (\tilde{\eta}_V\mathbb{E}[T^{-1}])^2$$

and

$$\left\|\begin{bmatrix}\hat{\boldsymbol{D}}_{i_{T,k}}\hat{\boldsymbol{D}}_{i_{T,k}}^\top & \boldsymbol{O}\\ \boldsymbol{O} & \hat{\boldsymbol{D}}_{i_{T,k}}^\top\hat{\boldsymbol{D}}_{i_{T,k}}\end{bmatrix}\right\|_2 \lesssim (\tilde{\eta}_V\mathbb{E}[T^{-1}])^2.$$

by Lemma 19. Then we obtain the error bound as

$$\frac{1}{\eta_{KQ}}\left\|\boldsymbol{W}_{KQ} - \boldsymbol{W}_{KQ}^*\right\|_2$$

$$\lesssim \left\| \sum_{T=5}^{L-1} \frac{M_{KQ}^T}{M_{KQ}} \frac{1}{M_{KQ}^T} \sum_{i_T} \left( \hat{C}_{i_T} - \mathbb{E}[\hat{C}_{i_T}|T] \right) \right\|_2 + \left\| \sum_{T=5}^{L-1} \left( \frac{M_{KQ}^T}{M_{KQ}} - q_{T(\ell)} \right) \mathbb{E}[\hat{C}_{i_T}|T] \right\|_2$$

$$+ \left\| \sum_{T=5}^{L-1} \frac{M_{KQ}^T}{M_{KQ}} \sum_{k=N_{\mathrm{trg}}+1}^{N} \frac{M_{KQ}^{T,k}}{M_{KQ}^T} \frac{1}{M_{KQ}^{T,k}} \sum_{i_{T,k}} \left( \hat{D}_{i_{T,k}} - \mathbb{E}[\hat{D}_{i_{T,k}}|T,k] \right) \right\|_2$$

$$+ \left\| \sum_{T=5}^{L-1} \frac{M_{KQ}^T}{M_{KQ}} \sum_{k=N_{\mathrm{trg}}+1}^{N} \left( \frac{M_{KQ}^{T,k}}{M_{KQ}^T} - \frac{1}{N - N_{\mathrm{trg}}} \right) \mathbb{E}[\hat{D}_{i_{T,k}}|T,k] \right\|_2$$

$$+ \left\| \sum_{T=5}^{L-1} \left( \frac{M_{KQ}^T}{M_{KQ}} - q_{T(\ell)} \right) \mathbb{E}[\hat{D}_{i_{T,k}}|T] \right\|_2 + \tilde{\eta}_V \mathbb{E}[T^{-1}] O_\infty(N_{\mathrm{trg}}/N)$$

$$\lesssim \tilde{\eta}_V \mathbb{E}[T^{-1}] \left( \sum_\ell \sqrt{q_\ell} \right) \sqrt{\frac{N \log\left( (L+N)\epsilon^{-1} \right)}{M_{KQ}}} + \tilde{\eta}_V \mathbb{E}[T^{-1}] O_\infty(N_{\mathrm{trg}}/N)$$

in the same way as bounding $\boldsymbol{W}_V$. We used $M_{KQ}^T \simeq q_{\ell(T)} M_{KQ}$, $M_{KQ}^{T,k} \simeq N^{-1} M_{KQ}^T$, (matrix-) Hoeffding's inequalities, and $\|\hat{C}_{i_T}\|_2, \|\hat{D}_{i_{T,k}}\|_2 \lesssim \tilde{\eta}_V \mathbb{E}[T^{-1}]$ by Lemma 19. $\qquad\square$

Based on these finite sample analyses, Theorem 6 can be obtained similarly to Theorem 5.

**Proof.** [Proof of Theorem 6] Note that, the approximation in Section B.2.1 still applies, if the finite-sample error with respect to $\boldsymbol{W}^{KQ}$ is falling into $\tilde{\eta} O(N_{\mathrm{trg}}/N)$. From Lemma 20, it suffices to set $M_{KQ} \gtrsim \mathrm{poly} \log N \cdot \frac{N^3}{N_{\mathrm{trg}}^2} \left( \sum_\ell \sqrt{q_\ell} \right)^2$. Together with the requirement $M_V \gtrsim \mathrm{poly} \log N \cdot \frac{N^5}{N_{\mathrm{trg}}^2} \left( \frac{\sum_{\ell \in \mathcal{S}} \sqrt{q_\ell}}{\sum_{\ell \in \mathcal{S}} q_\ell \ell^{-1}} \right)^2$ in Corollary 18 we obtain the assertion. $\qquad\square$

# D  Proof of Proposition 8

We first show that

$$\boldsymbol{q}^* = (q_1^*, q_2^*, \ldots, q_U^*) = \frac{(1, 2, \ldots, N_{\mathrm{trg}}, 0, \ldots, 0)}{Z} \quad (Z = \frac{N_{\mathrm{trg}}(N_{\mathrm{trg}} + 1)}{2})$$

satisfies the KKT condition of the LP

$$\mathbb{P} : \begin{cases} \text{minimize} & \sum_{\ell=1}^{U} q_\ell \ell^2 \\ \text{subject to} & \frac{\max_{\ell=1}^{U} q_\ell \ell^{-1}}{\sum_{\ell=1}^{U} q_\ell \ell^{-1}} \leq N_{\mathrm{trg}}^{-1} \\ & \sum_{\ell=1}^{U} q_\ell = 1 \\ & q_1, \ldots, q_U \geq 0 \end{cases},$$

and show its uniqueness.

The KKT condition of $\mathbb{P}$ is

$$\ell^2 + (\lambda_\ell - \sum_{\ell'=1}^{U} N_{\mathrm{trg}}^{-1} \lambda_{\ell'}) \ell^{-1} - \mu_\ell + \nu = 0 \quad (\ell \in [U]), \tag{D.1}$$

$$\lambda_\ell (q_\ell \ell^{-1} - N_{\mathrm{trg}}^{-1} \sum_{\ell'} q_{\ell'} (\ell')^{-1}) = 0 \quad (\ell \in [U]), \tag{D.2}$$

$$\mu_\ell(-q_\ell) = 0 \quad (\ell \in [U]), \tag{D.3}$$

$$q_\ell \ell^{-1} \leq N_{\mathrm{trg}}^{-1} \sum_{\ell'=1}^{U} q_{\ell'} \ell'^{-1} \quad (\ell \in [U]), \tag{D.4}$$

$$q_1 + \ldots + q_U = 1 , \tag{D.5}$$

$$\lambda_\ell \geq 0, \mu_\ell \geq 0 \quad (\ell \in [U]). \tag{D.6}$$

We construct $(\boldsymbol{\lambda} = \{\lambda_\ell\}, \boldsymbol{\mu} = \{\mu_\ell\}, \nu)$ such that $(\boldsymbol{q}, \boldsymbol{\lambda}, \boldsymbol{\mu}, \nu)$ satisfies these conditions: by construction, (D.4) and (D.5) are already satisfied. Here, note that

$$q_\ell \ell^{-1} = \begin{cases} Z^{-1} & (\ell \leq N_{\text{trg}}), \\ 0 & (\ell > N_{\text{trg}}). \end{cases}$$

Thus, from (D.2) and (D.3) we have $\lambda_\ell = 0$ $(\ell > N_{\text{trg}})$ and $\mu_\ell = 0$ $(\ell \leq N_{\text{trg}})$.

Now it remains to satisfy (D.1), not braking (D.6). For $\ell \in [N_{\text{trg}}]$, (D.1) reduces to the following linear equations:

$$\left( \boldsymbol{I}_{N_{\text{trg}}} - \begin{bmatrix} N_{\text{trg}}^{-1} & \cdots & N_{\text{trg}}^{-1} \\ \vdots & \ddots & \vdots \\ N_{\text{trg}}^{-1} & \cdots & N_{\text{trg}}^{-1} \end{bmatrix} \right) \begin{bmatrix} \lambda_1 \\ \vdots \\ \lambda_{N_{\text{trg}}} \end{bmatrix} = - \begin{bmatrix} 1^3 + \nu \cdot 1 \\ \vdots \\ N_{\text{trg}}^3 + \nu \cdot N_{\text{trg}} \end{bmatrix}.$$

Noting that the sum of the all entries in the vector obtained by evaluating left-hand side is zero, we obtain

$$\nu = -\frac{1 + \cdots + N_{\text{trg}}^3}{1 + \cdots + N_{\text{trg}}} = \frac{N_{\text{trg}}(N_{\text{trg}} + 1)}{2}.$$

We can also observe $\lambda_1 \geq \cdots \geq \lambda_{N_{\text{trg}}}$ since the right-hand side is decreasing w.r.t the vector index.

Since $\boldsymbol{w} = [1, 1, \ldots, 1]^\top$ belongs to the right kernel of $\left( \boldsymbol{I}_{N_{\text{trg}}} - \begin{bmatrix} N_{\text{trg}}^{-1} & \cdots & N_{\text{trg}}^{-1} \\ \vdots & \ddots & \vdots \\ N_{\text{trg}}^{-1} & \cdots & N_{\text{trg}}^{-1} \end{bmatrix} \right)$, we can

shift $\boldsymbol{\lambda}$ by this vector to ensure $\lambda_{N_{\text{trg}}} = 1$ (then $\boldsymbol{\lambda} \geq 0$), meaning

$$1 - \bar{\lambda} = -N_{\text{trg}}^3 - \nu \cdot N_{\text{trg}} \Leftrightarrow \bar{\lambda} = \frac{1}{2} N_{\text{trg}}^3 - \frac{1}{2} N_{\text{trg}}^2 + 1$$

where $\bar{\lambda} = N_{\text{trg}}^{-1} \sum_{\ell=1}^{N_{\text{trg}}} \lambda_\ell$. We now consider (D.1) with $\ell > N_{\text{trg}}$. Since

$$\mu_\ell = \nu + \ell^2 - \ell^{-1} \bar{\lambda}$$
$$\geq (N_{\text{trg}} + 1)^2 - \frac{N_{\text{trg}}(N_{\text{trg}} + 1)}{2} - \frac{1}{2} N_{\text{trg}}^2 + \frac{1}{2} N_{\text{trg}} - \frac{1}{N_{\text{trg}}} \quad = 2N_{\text{trg}} + 1 - \frac{1}{N_{\text{trg}}} \geq 1,$$

and then we can now determine $\boldsymbol{\mu}$ satisfying (D.6). Therefore, obtained $(\boldsymbol{q}^*, \boldsymbol{\lambda}, \boldsymbol{\mu}, \nu)$ satisfies the KKT condition.

From [Man79], to show the uniqueness it suffices to show that for any $\boldsymbol{p} \in \mathbb{R}^U$, there exists $\epsilon > 0$ such that even if we replace the objective function to $\sum_{\ell \in [U]} (\ell^2 + \epsilon p_\ell) q_\ell$, $\boldsymbol{q}^*$ is optimal. We can easily see this by reconsidering KKT condition—while the only effect by changing the objective is the nonnegativeness of $\boldsymbol{\lambda}$ and $\boldsymbol{\mu}$ (D.6), these parameters are continuous with respect to the perturbation, and we can still ensure nonnegativeness since for $\boldsymbol{q} = \boldsymbol{0}$ we already obtained positive parameters.

# E    Detailed Experimental Settings and Results

## E.1    Detailed Settings for Section 4.2

We introduce the detailed settings for the full-traning experiment.

**Architecture.**

- **Embedding.** We use embeddings obtained by concatenating the positional embedding and the token embedding, i.e., $\begin{bmatrix} \boldsymbol{p}_t \\ \boldsymbol{e}_{z_t} \end{bmatrix}$ where $\boldsymbol{p}_t$ and $\boldsymbol{e}_{z_t}$ are one-hot vectors with ones at $t$-th and $z_t$-th entries, respectively. The previous-token embedding in (2.2) is omitted.

- **Transformer blocks.** Each layer consists of a single-head attention module with separate Key-Query matrices, a GeLU-based MLP, and residual connections:

$$\boldsymbol{x}_t \leftarrow \boldsymbol{x}_t + \mathrm{MLP}\big(\boldsymbol{x}_t + \boldsymbol{W}_V \boldsymbol{X}_{1:t} \mathrm{Softmax}(\boldsymbol{X}_{1:t}^\top \boldsymbol{W}_K^\top \boldsymbol{W}_Q \boldsymbol{x}_t)\big),$$

where

$$\mathrm{MLP}(\boldsymbol{x}) = \boldsymbol{W}_{\mathrm{MLP},2} \mathrm{GeLU}(\boldsymbol{W}_{\mathrm{MLP},1}\boldsymbol{x} + \boldsymbol{b}_1) + \boldsymbol{b}_2.$$

Three such layers are stacked, followed by a linear projection of size $(N, D)$ that maps the final embeddings (dimension $D$) to the vocabulary of size $N$. We initialized $\boldsymbol{W}_K, \boldsymbol{W}_Q, \boldsymbol{W}_V, \boldsymbol{W}_{\mathrm{MLP},1}$ and $\boldsymbol{W}_{\mathrm{MLP},2}$ using Xavier initialization [GB10], while biases $\boldsymbol{b}_1$ and $\boldsymbol{b}_2$ are initialized from the zero vector. The size of $\boldsymbol{W}_{\mathrm{MLP},1}$ and $\boldsymbol{W}_{\mathrm{MLP},2}$ are $(4D, D)$ and $(D, 4D)$, respectively, where $D = N + L$ is the embedding dimension. The transformed embedding at the last layer is fed into the trainable linear output layer $\boldsymbol{W}^O$ of size $(N, D)$, initialized using Xavier, before softmax.

**Training.** Training was performed using AdamW with both the learning rate and weight decay set to $10^{-2}$, using 32,768 training samples. We prepared 1,024 in-distribution samples drawn from the same distribution as the training data and stopped training once the accuracy exceeded 90% on these samples.

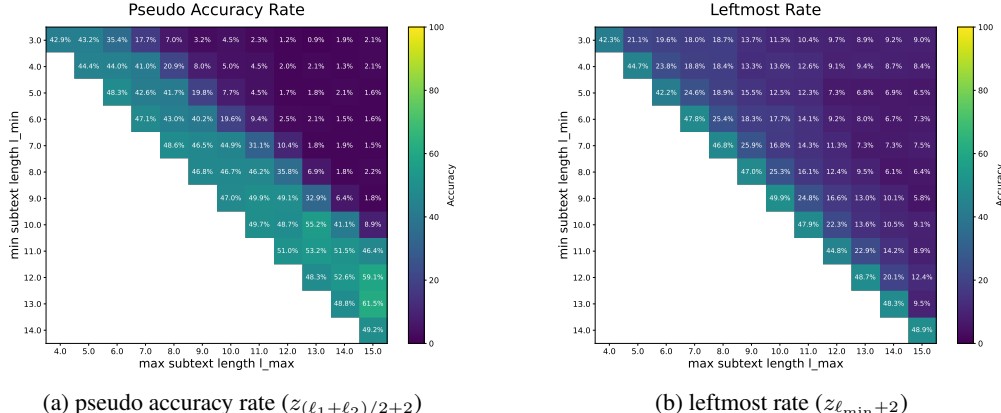

(a) pseudo accuracy rate ($z_{(\ell_1+\ell_2)/2+2}$)  (b) leftmost rate ($z_{\ell_{\min}+2}$)

Figure 5: Map of two types of errors due to the positional shortcut. Note that both errors can probabilistically coincide with the correct answer, and such cases are not excluded.

