# OpenReview forum: "From Shortcut to Induction Head: How Data Diversity Shapes Algorithm Selection in Transformers"
_NeurIPS.cc/2025/Conference — NeurIPS 2025 spotlight_

### Official Review · Reviewer_eKuv · 2025-07-03

**Clarity:** 4
**Significance:** 3
**Originality:** 4
**Rating:** 5
**Confidence:** 3

**Summary:**

This paper provides a theoretical analysis of how pretraining data diversity influences whether a transformer learns a generalizable induction head or a positional shortcut. Focusing on a minimal trigger-copying task, the authors show a phase transition governed by a max-sum ratio, which quantifies data diversity. With high-diversity data (low ratio), the model learns the induction head, while low-diversity data leads to positional memorization that fails on out-of-distribution inputs. The analysis is first carried out under infinite sample size and then extended to the finite-sample regime. The latter yields sample complexity bounds and identifies an optimal data distribution that balances generalization and statistical efficiency. Synthetic experiments on a two-layer transformer validate the theoretical prediction that increasing data diversity robustly induces OOD generalization.

**Questions:**

1. **Impact of Multiple Trigger Tokens:** "Your analysis assumes a single, fixed trigger token (`e_1`). What happens if the trigger is instead drawn from a set of `K` possible tokens? This extension intuitively seems to increase the learning complexity of the induction head, requiring `W_KQ` to learn a more general identity-matching function (a sub-identity matrix) rather than a single association. How would this affect two of your core results:
	- **The Transition Point:** Would the `R(D_l)` threshold required to learn the induction head become stricter?
	- **The Sample Complexity:** How would the sample complexity bound in Theorem 7 change? Would it now need to scale with the number of possible triggers `K`?"

2.  **Formalizing the Max-Sum Ratio's Role:** "The core of your paper's theoretical results is that the max-sum ratio, `R(D_l)`, determines whether the model learns a positional shortcut or an induction head. This is intuitively explained as a competition between the two mechanisms.

	Would it be possible to formalize this intuition by directly mapping the terms of the ratio to the attention score?

	Specifically, using the attention score `s_t` from the proof of Proposition 5, can you show that:
	- The effective magnitude of the positional shortcut component of `s_t` is proportional to the numerator of the ratio, `max_l q_l l^{-1}`?
	- The magnitude of the induction head component of `s_t` is proportional to the denominator, `\sum_l q_l l^{-1}`?

	This would provide a clear, formal bridge between the `R(D_l)` criterion and the final learned attention scores.

3. The experiments show a sharp drop in OOD loss as K increases, but do the empirical transition points align quantitatively with your predicted $\epsilon_1$,$ \epsilon_2$?

**Ethical Concerns:**

["NO or VERY MINOR ethics concerns only"]

**Final Justification:**

I recommend the acceptance of this paper. The discussion simply served to confirm my already positive evaluation.

- Resolved Issues: The points I raised in my review were not critical flaws but rather clarification on the extensibility and formal underpinnings of the work. The authors' responses were satisfactory, providing intuition for the multi-trigger case while confirming the role of the max-sum ratio and the model's alignment with experiments.

- Unresolved Issues: There are no unresolved issues.

- Final Recommendation: I find the main question studied in the paper interesting, and the theoretical framework is solid. The discussion did not fundamentally change my assessment but rather reinforced it.

**Limitations:**

yes

**Quality:**

3

**Strengths And Weaknesses:**

**Strengths:**

1.  **Theoretical Framework:** This paper addresses an interesting and timely question with clarity and rigor. The authors introduce the "max-sum ratio", a novel and mathematically well-defined quantity that enables rigorous study of how data diversity influences algorithm selection in transformer models when solving in-context associative recall tasks. This provides a concrete, predictive framework for analyzing training dynamics in transformers.
2. **Analysis:** The analysis of the phase transition in both infinite and finite-sample settings is a significant theoretical contribution.
3.  **Clear Mechanistic Story:** The manuscript is well written, and I enjoy reading it. It provides a clear, guided explanation of how positional signals "diffuse" with diverse data while content-based signals "concentrate", leading to the emergence of one mechanism over the other. This is supported by detailed proofs and validated by experiments.

**Weaknesses:**

1.  **Simplifying Assumptions:** The analysis relies on key simplifications.  The input embedding `x_t` is hard-coded with previous-token information (`e_{z_{t-1}}`), meaning the model is given the feature required for induction rather than learning to construct it. This sidesteps the full complexity of how induction heads form in standard two-layer models, but it is also fairly standard in related works.

2.  **Focus on a Single, Special Trigger:** The analysis is restricted to a single, fixed trigger token. This simplifies the induction mechanism to a single association rather than the more general identity-matching function (`W_KQ` learning a sub-identity matrix) required for real-world induction heads that operate on arbitrary token patterns.

3.  **Single-Step Gradient Analysis:** The results are derived from a single gradient step from a zero initialization. This approximation nicely captures the investigated mechanisms but may not fully represent the complex dynamics of a full training run with an adaptive optimizer.

---

> ### Author Rebuttal · Authors · 2025-07-31
>
> We thank the reviewer for the helpful feedback. We address the comments and questions below.
>
> ## Impact of multiple trigger tokens
> Extending the analysis to the case of multiple triggers is an interesting and natural direction, and we are actively working on this extension.
>  The key idea behind our current analysis is that the relative strength of two peaks in the attention key-query matrix—namely, the positional memory and the induction head components corresponding to terms (a) and (b) in Equation (3.1)—determines which of the two algorithms is implemented by the transformer. Our results suggest that when the pretraining dataset (i.e., the positions of the triggers) is diverse, the peak associated with the positional memory becomes dispersed across multiple trigger positions, thereby making the induction head dominant.
>
> In contrast, when there are multiple triggers—drawn from a set of $K$ tokens, as you suggested—the induction head signal, which depends on the semantics of the triggers, also becomes dispersed. This yields two implications:
> - **Transition Point**. Since multiple triggers make the induction head more difficult to implement, we expect that a higher level of diversity in the pretraining data is required to induce a phase transition from positional memorization to the induction head regime.
> - **Sample Complexity**. As the induction head signal is attenuated due to the semantic dispersion across multiple triggers, the number of pretraining samples required to prevent the signal from being buried in noise—in other words, the sample complexity required for implementing the induction head—is expected to increase and dependent on $K$.
>
> We believe these implications have the potential to make the transition point more complex but to provide a rich characterization of the competition between positional and semantic signals.
>
> ## Formalizing the Max-Sum Ratio's Role
>
> As you pointed out, the effective magnitude of the positional shortcut and that of the induction head component in $s_t (= x_{t}^\top W_{KQ} x_{T})$ are proportional to the numerator and the denominator, respectively. In short, the max-sum ratio governs the relative strength of the positional shortcut and the induction head in the attention logits. In light of your comment, we will discuss this intuition in the main text.
>
> ## On the experiment
>
> In practice, the precise location of the transition point can depend on architectural and embedding-specific details, and thus cannot be fully predicted in a quantitative manner based solely on the max-sum ratio.
>
> Being said, some qualitative insights can still be obtained.   For instance, if we consider a uniform distribution over an interval of width $K$, namely $[\ell_0, \ell_0 + 1, \ldots, \ell_0 + K - 1]$, and we shift this distribution to the right (i.e., toward longer contexts) while keeping the total width $K$ fixed, the max-sum ratio decreases. As suggested by Theorem 7, a smaller max-sum ratio is favorable for OOD generalization. Therefore, shifting the distribution to the right reduces the required width $K$ for achieving OOD generalization.　This is consistent with the experimental result.
>
> ## Simplified setup
> We provide the following remark concerning our simplified setup:
> - **Simplified Embedding.** Our simplified embedding scheme is intended to model the behavior in deeper architectures, where information from previous tokens flows into the residual stream [1].  While the analysis may become more involved, a simplified multi-layer architecture equipped with residual connections and relative positional embeddings [2] would reproduce our theoretical insights.
> - **Single-step analysis.** We remark that our learning setup — based on one gradient descent step with a large learning rate — is intended to model the behavior of parameters escaping from saddle points during the initial phase of gradient-based algorithms. This setup has been extensively studied in the neural network theory literature, and in many feature learning settings it is known that alternative gradient-based training algorithms such as one-pass SGD yield similar statistical efficiency [3,4].
>
> [1] Bietti et al. Birth of a Transformer: A Memory Viewpoint.
> [2] Wang and Sato.  Rethinking Associative Memory Mechanism in Induction Head.
> [3] Abbe, Boix-Adsera, and Misiakiewicz: SGD learning on neural networks: leap complexity and saddle-to-saddle dynamics.
> [4] Barak et al: Hidden Progress in Deep Learning: SGD Learns Parities Near the Computational Limit.
>
> We would be happy to answer any follow-up questions in the discussion period.

---

> > ### Comment · Reviewer_eKuv · 2025-08-07
> >
> > Thank you for the detailed rebuttal and the insightful discussion. Your responses have addressed my questions and clarified the points I raised. I am happy to confirm my positive evaluation and recommend the paper for acceptance.

---

### Official Review · Reviewer_DDFp · 2025-07-03

**Clarity:** 3
**Significance:** 3
**Originality:** 3
**Rating:** 5
**Confidence:** 4

**Summary:**

This paper studies how the diversity of pretraining data influences whether a Transformer learns a generalizable algorithm (such as an induction head) or relies on a brittle, position-specific shortcut. Using a simplified trigger-output copying task, the authors analyze a single-layer Transformer trained with one step of gradient descent. A key contribution is the identification of a sharp transition in the model’s behavior, controlled by a metric they introduce—the “max-sum ratio”—which captures the diversity of the training distribution. When this ratio exceeds a certain threshold, the model shifts from positional memorization to learning an induction-like mechanism that generalizes to unseen data. The theoretical findings are further supported by experiments on a two-layer model with relative positional encoding, which exhibit the same phase transition and corresponding improvement in out-of-distribution performance.

**Questions:**

I really enjoyed the theoretical framing in this paper—especially how it connects data diversity to the emergence of associative recall. That said, I’m curious about how these insights might carry over to more realistic or complex settings. For example, challenges like long-context retrieval (e.g., in “Needle in a Haystack” or multi-key recall) remain open. Do you think the max-sum ratio you introduce could be used to inform how we design synthetic pretraining datasets to better support such abilities?

Also, I wonder how your theory might generalize to hybrid architectures like Transformer-Mamba. These models often show different recall failure modes, possibly due to the limited state size in the recurrent component. Do you have any thoughts on how your framework could help us even mitigate these limitations?

**Ethical Concerns:**

["NO or VERY MINOR ethics concerns only"]

**Final Justification:**

I maintain my positive score

**Limitations:**

yes

**Quality:**

3

**Strengths And Weaknesses:**

# Strength

This paper gives a sharp and intuitive take on how data diversity influences what kinds of algorithms Transformers end up learning. I found the idea of "positional memory" especially compelling—it neatly explains why models often struggle to generalize when the input structure shifts. The theoretical results are solid, especially the phase transition between learning a positional shortcut vs. a more general induction-based solution. I also appreciated the experimental setup: validating the theory on a stronger model with relative position encodings really drives home that the insights aren't just artifacts of toy settings.


# Weakness

I do not have major concerns about this work. My feedback consists of some minor points.

1. The theoretical tools used are standard, and the gradient-based perspective has been widely adopted in the study of in-context learning. While the absolute embedding is arguably restrictive, it serves well for building intuition. From this perspective, the theoretical results are not surprising, though I appreciate the authors' effort in formalizing them.

2. The provided theoretical analysis cannot subsume multiplicative positional encodings like RoPE. Although the authors argue that their findings should generalize to relative positional encoding schemes (given their setup where the relative distance is also fixed), and the experimental results appear to support this claim, this extension lacks a fundamental theoretical justification. Understanding how RoPE-equipped models might form such positional memories is a highly non-trivial research question worthy of further investigation. It would be valuable to build a dedicated theoretical framework for this case.

---

> ### Author Rebuttal · Authors · 2025-07-31
>
> We thank the reviewer for the helpful feedback. We address the comments and questions below.
>
> ### Implications of the max-sum ratio for realistic settings
>
> While the max-sum ratio is derived under our simplified theoretical setup, we believe it provides a high-level guideline for designing datasets that promote better generalization.
>
> Primarily, the max-sum ratio encourages diversity in the task distribution, as discussed in the main text.  Further intuition behind the max-sum ratio is that the task distribution, when reweighted by the inverse of the context length (i.e., $q_\ell$ weighted by $\ell^{-1}$), should be approximately uniform in order for the max-sum ratio to attain its minimum.  In other words, $q_\ell$ associated with smaller $\ell$—which corresponds to a task where positional information can be memorized easily—should take relatively smaller values.
>  Accordingly, the max-sum ratio suggests the following simple high-level principles for dataset design: (i) selecting a more diverse range of tasks, and (ii) placing greater emphasis on tasks in which positional signals are harder to obtain.
>
> ### Architectural and Embedding Considerations
> Thank you for the suggestion.  Indeed, it is challenging to straightforwardly extend our theoretical framework to multiplicative embeddings like RoPE or to SSMs that depart from the transformer architecture with absolute embedding.
> That said, the high-level insights regarding dataset design discussed in the previous section—such as promoting some notion of diversity in the pretraining data distirbution and emphasizing tasks where positional signal is harder to obtain—are not specific to any particular architecture or embedding scheme. As such, we believe it is worthwhile to explore whether these dataset design principles remain effective in these more advanced settings.
>
> A step toward the RoPE setting is to incorporate an additional layer. While the analysis may become more involved, the use of a residual stream potentially allows us to extend our one-layer architecture to a multi-layer architecture equipped with relative positional embeddings [1,2].
> Although further extensions would be required to fully capture the multiplicative structure of RoPE, we view this as a meaningful future step in that direction.
>
> [1] Bietti et al. Birth of a Transformer: A Memory Viewpoint.
> [2] Wang and Sato.  Rethinking Associative Memory Mechanism in Induction Head.
>
> We would be happy to answer any follow-up questions in the discussion period.

---

> > ### Comment · Reviewer_DDFp · 2025-08-06
> >
> > Thank you for the pointers. I have no further questions and keep my positive score. Good luck :)

---

### Official Review · Reviewer_tEKW · 2025-07-03

**Clarity:** 2
**Significance:** 2
**Originality:** 2
**Rating:** 3
**Confidence:** 2

**Summary:**

The in-context learning behavior of transformer-backed large language models is known to be implemented through induction heads. This paper aims to study how the pre-training data distribution and composition influence shaping of induction heads and other attention mechanisms. The authors first examine how gradient-based training works on shallow, single-layer transformers in the infinite data regime and show that pre-training data distribution does indeed "affect the models' learned algorithms." Then, the authors theoretically derive the threshold in data diversity that instigates the model to transition from shortcut learning to the development of induction heads. Lastly, the authors empirically show that the models trained on a wider variety of context length distributions show stronger OOD generalization performance than those trained on a narrow scope of data to further support their theoretical findings.

**Questions:**

Please refer to the weaknesses section.

**Ethical Concerns:**

["NO or VERY MINOR ethics concerns only"]

**Final Justification:**

I have read the authors' clarification regarding my questions. My concerns are partially resolved, and thus I am raising my score to borderline reject. Given my limited expertise in this area, I kindly ask AC to weigh the other reviewer's judgment more when making the final decision.

**Limitations:**

Yes

**Paper Formatting Concerns:**

References are cited and formatted incorrectly.

**Quality:**

2

**Strengths And Weaknesses:**

> **Strengths**

- Understanding how transformers develop induction heads and other attention mechanisms that eventually lead to the emergent behaviors in LLMs is an important research topic that can advance our understanding of AI.

- The authors' findings regarding the relationship between data diversity and OOD generalization are technically sound and well-aligned with common beliefs in machine learning.

- Albeit minimal, the authors do make an attempt at providing empirical evidence for their findings.

> **Weaknesses**

- I do not quite grasp the authors initial statement and assumption on the transformers' inability to perform OOD length generalization: "despite their capacity for in-context learning, models often fail to generalize out-of-distribution." To the best of my understanding, in-context learning allows LLMs, transformer-backed architectures, to learn new OOD tasks through in-context description and examples (as suggested in a seminal paper "language models are few-shot learners.") Although LLMs do have context limit, but isn't that due to their limited context length? If so, I am not sure whether I can agree with the above statement especially in the era of LLMs with a million+ tokens worth of context length and multi-agent LLM systems that can handle multi-turn long-horizon tasks.

- Although I concur with the authors on that developing theoretical frameworks to understand the inner-workings of transformers is important, I remain quite doubtful of the impact of the findings and insights provided in this paper. There are several papers that report the importance of pre-training data on LLMs' performance and investigate how to optimize the pre-training data recipe to improve their downstream performance on various tasks [1, 2, 3]. I have only cited a few papers here, but the related works sections of these cited papers discuss more works. If so, what additional insight can a theoretical analysis on a single-layer transformer contribute to the LLML community?

[1] DataMan: Data Manager for Pre-training Large Language Models

[2] Deciphering the Impact of Pretraining Data on Large Language Models through Machine Unlearning

[3] Rewriting Pre-Training Data Boosts LLM Performance in Math and Code

[4] Investigating the Pre-Training Dynamics of In-Context Learning: Task Recognition vs. Task Learning -> not exactly on data distribution but studies how pre-training affects in-context learning.

- The numerical experiment section is quite hastily written, and many of the necessary details for interpreting the results are missing. In particular, the data generation process was difficult for me to follow. I think this section could offer more details, given that NeurIPS now allows up to 9 nine content pages, and the current version only has 8 (and a few more lines).
   - According to Definition 1, $N$ denotes the vocabulary size and $L$ denotes the maximum sequence length. What vocabulary corpus did the authors use? What are the values of $N$ and $L$ for this dataset?
   - How is the token 1 designed as the trigger token? Do we treat it as a special token like <EOS> with an XML-style tag?

---

> ### Author Rebuttal · Authors · 2025-07-31
>
> We thank the reviewer for the helpful feedback. We address the comments and questions below.
>
> ## OOD length generalization
> > I do not quite grasp the authors initial statement and assumption on the transformers' inability to…
>
> Thank you for raising this point. While LLMs tend to work well even with very long contexts, they sometimes rely on heuristic or spurious features for solving certain tasks, which can manifest itself in a failure to generalize to certain OOD scenarios. These failures to generalize have been observed, e.g., in linguistic tasks (e.g. [1]), or in simple algorithmic tasks requiring length generalization (e.g. [2,3]). Generally, these tasks can be much more complicated than simple few-shot learning, and often involve the composition of small algorithmic primitives that need to be robust to changes in their inputs.
>
> [1] McCoy, Pavlick and Linzen.  Right for the Wrong Reasons: Diagnosing Syntactic Heuristics in Natural Language Inference.
>
> [2] Anil et al. Exploring Length Generalization in Large Language Models.
>
> [3] Zhou et al. What Algorithms can Transformers Learn? A Study in Length Generalization.
>
> Our setup describes possibly the simplest algorithmic scenario in a transformer that is prone to such failures, namely copying a token based on heuristic positional information instead of learning a general copy mechanism that works regardless of position. The simplicity of our task allows us to obtain a precise characterization of the phenomena at play, such as how the choice of data distribution affects gradient dynamics, and how much diversity is needed in order to properly avoid the heuristic “shortcut” solutions and obtain reliable models. Our study paves the way towards understanding more complex scenarios which may involve compositions of such small algorithmic components.
>
> ## Value of theoretical works on transformers
> > Although I concur with the authors on that developing theoretical frameworks…
>
> This is a good question. While it is clear at this point that the community has realized the importance of pre-training data and data efficiency, many important questions remain open in this area, and the potential to further improve data efficiency could unlock significant improvements in model performance, as well as cost savings in pre-training. This is even more the case when you start considering synthetic / simulated data, instead of filtering existing available data, as in some of the referenced works. An important long-term outcome that we’re aiming towards is a way to find the smallest possible dataset that would allow us to train a model that perfectly solves a range of complex algorithmic reasoning tasks. Even just formalizing this question is very difficult. Our work takes a first step by considering possibly the simplest algorithmic task (copying with induction heads), and trying to understand how the data distribution affects the learning dynamics, and to characterize how diversity leads to a transition from using spurious positional heuristics to generalizable copy mechanisms. While the task in itself is quite basic, you could imagine extensions of our work that compose many such small algorithmic components in a larger transformer in a reliable way to achieve more complex algorithmic tasks. We hope this clarifies our motivation, and the significance of this initial step towards a much grander goal.
>
> ## On the numerical experiment
>
> Thank you for the feedback.  We will elaborate on the experimental setting in the revision.
>
> - **Data Model.** The data used in the experiment of Figure 1 is synthetic data following the data-generating process in Definition 1. In particular,
> We set the vocabulary size to $N = 36$ and maximum sequence length to $L = 36$.
> The vocabulary itself consists of natural numbers $\{1, 2, \dots, N=36\}$; each sample sequence is of length up to $L = 36$, such as $[10, 8, 1, 4, 5, 8, 1, \dots]$, and the probability distribution over data sequences is defined in Definition 1.
> We do not treat token 1 as a special token; rather, we embed it in the same manner as any other token, as described below (This is because it is required to construct the induction head without explicitly signaling that token 1 carries a different role from the other $N-1$ tokens.)
> - **Embeddings.** Each token $ z_i \in [N]$ (where $i \in [L]$) is represented by the concatenation of a zero vector of length $L$ and a one-hot vector of length $N$ whose $z_i$-th component is one. For positional embeddings, we use the concatenation of a one-hot vector of length $L$ (with one at the $i$-th position) and a zero vector of length $N$. See also Appendix E.1 for further details.
> - **Architecture and Training.** Based on the above, we train a two-layer architecture as described in Section 4. We perform optimization using the Adam optimizer with a learning rate of 0.0001, with early stopping every 250 steps.  We plan to integrate the explanations of architectural and training details, which are currently split between Section 4 and Appendix E.1.
>
> We would be happy to answer any follow-up questions in the discussion period.

---

> > ### Comment · Reviewer_tEKW · 2025-08-05
> >
> > I have read through authors' clarification regarding my questions.
> >
> > I appreciate that the authors provided further discussion on the contributions of their work, as well as clarifying experimental details. My concerns are partially resolved, and thus I am raising my score to borderline reject. Given my limited expertise in this area, I kindly ask AC to weigh the other reviewer's judgment more when making the final decision.

---

### Official Review · Reviewer_pSCb · 2025-07-06

**Clarity:** 2
**Significance:** 2
**Originality:** 3
**Rating:** 3
**Confidence:** 3

**Summary:**

This paper investigates how the diversity of pretraining data influences the mechanism selected by transformers, particularly whether they adopt robust induction heads or brittle positional memorization. The authors formalize a minimal synthetic task and provide a rigorous theoretical and empirical analysis. They define a max-sum ratio to quantify pretraining diversity, prove a phase transition in mechanism selection, and derive optimal sampling strategies for data. Controlled experiments validate the theory.

**Questions:**

- Can the authors clarify how the current analysis might extend to capture the full training dynamics?
- How do the authors envision the extension of their results to more realistic pretraining settings, such as natural language corpora? Can the current theory predict induction head behavior in standard language modeling benchmarks?
- To what extent do the authors believe the layerwise training assumption affects their main theoretical conclusions?
- To what extent does the simplification of the quadratic sturcture in attention affect the overall theoretical conclusions?

**Ethical Concerns:**

["NO or VERY MINOR ethics concerns only"]

**Final Justification:**

Although other concerns have been addressed in the rebuttal, two critical issues remain unresolved: (1) the theoretical analysis is incomplete; (2) the relevance to realistic settings remains unclear. I will therefore maintain my current score.

**Limitations:**

See Weaknesses.

**Quality:**

3

**Strengths And Weaknesses:**

**Strengths.**
- The paper focus on a fundamental theoretical and practical question: how pretraining data influences mechanism selection in transformers.
- It introduces a principled criterion, the max-sum ratio,that characterizes the phase transition between positional shortcuts and induction heads.
- Gradient-based analysis provides clear insights into how different training regimes shape model behavior.


**Weaknesses.**
- The theoretical analysis is incomplete; it does not capture the full training dynamics across time.
- The synthetic trigger-output task is overly simplified and may not reflect structural properties of natural language. It remains unclear how the proposed theory and experiments relate to induction heads in real-world language modeling.
- The assumption of layerwise training, though common in recent works [1–3], may significantly affect the validity of the theoretical conclusions.
- The theoretical model (changing $W_K$ and $W_Q$ to $W_{KQ}$) simplifies the attention mechanism. A recent work [3] suggests that the quadratic form $W_KW_Q$ is crucial for the emergence of induction head. (Note that while the two forms are equivalent in terms of expressivity, their training dynamics are not equivalent.)


[1] Chen et al, 2024. Unveiling induction heads: Provable training dynamics and feature learning in transformers.
[2] Nichani et al, 2024. How transformers learn causal structure with gradient descent.
[3] Wang et al, 2024. How Transformers Get Rich: Approximation and Dynamics Analysis.

---

> ### Author Rebuttal · Authors · 2025-07-31
>
> We thank the reviewer for the helpful feedback. We address the comments and questions below.
>
> ## Extension to the full training dynamics and simultaneous training on the all layers
> We remark that our learning setup — based on one gradient descent step with a large learning rate — is intended to model the behavior of parameters escaping from saddle points during the initial phase of gradient-based algorithms. This setup has been extensively studied in the neural network theory literature, and in many feature learning settings it is known that alternative gradient-based training algorithms such as one-pass SGD yield similar statistical efficiency [1,2].
>
> Moreover, the experiments described in Section 4 were conducted using a multi-step Adam optimizer, where all layers were trained jointly. This suggests that the transition from memorization to out-of-distribution generalization is not merely an artifact of the one-step gradient descent setting assumed in our theory. We will include a more thorough empirical analysis of how the phenomena we identified unfold in multi-step optimization, including an in-depth examination of inner mechanisms such as attention patterns.
>
> [1] Abbe, Boix-Adsera, and Misiakiewicz: SGD learning on neural networks: leap complexity and saddle-to-saddle dynamics.
>
> [2] Barak et al: Hidden Progress in Deep Learning: SGD Learns Parities Near the Computational Limit.
>
> ## Insights on the realistic NLP settings
>
> Thanks for the question. Our goal is to provide a first step towards understanding how to design the most efficient data distributions for learning a range of algorithmic tasks in LLMs. While our focus is on a simple synthetic task involving copying, you could imagine scaling this up to much more involved algorithmic tasks with synthetic data, and possibly turning them into natural language reasoning tasks with the help of LLMs, something which has been recently successful (e.g., [3]). Indeed, while the copying operation is just a tiny step towards this grander goal, copying is plausibly an important building block in many more complex algorithms, and failing to have a reliable copy mechanism would easily cause the larger tasks to also fail. We hope that our precise characterization of how data affects training dynamics, and how diversity helps the model transition from relying on spurious (positional) heuristics to generalizable solutions, will help to study data efficiency of more complex tasks consisting of multiple such components.
>
> [3] Gunasekar et al.  Textbooks Are All You Need.
>
> ## Merged key and query matrices
>
> In our theoretical analysis, we apply a simplification that ties $W^K$ and $W^Q$ in our model, yet we still observe the emergence of induction heads. This suggests that a factorized form is not strictly necessary for induction heads to appear (although they may appear at different speeds in the two models due to different dynamics and implicit biases, as studied in the referenced paper “How Transformers Get Rich”). On the other hand, the experiments described in Section 4 use a setting where keys and queries are separated.  Taken together, these findings indicate that whether keys and queries are tied or separated is not essential for reproducing our theoretical insight.
>
> We would be happy to answer any follow-up questions in the discussion period.

---

> ### Comment · Reviewer_pSCb · 2025-08-07
>
> I appreciate the authors' detailed explanations. However, my main concerns remain unresolved:
>
> - Theoretical Analysis: The analysis remains incomplete. As acknowledged in the rebuttal, the paper examines only the dynamics of a single gradient step, without addressing the full training trajectory. While some theoretical works focus on the first step under simplified assumptions, they typically also establish global convergence—something this paper does not.
>
> - Relevance to realistic settings: Although this is a theoretical paper, its data-centric viewpoint is interesting. Nonetheless, it is important to clarify its relevance to realistic NLP scenarios. At least, the paper should discuss the connection between the studied task and standard NLP tasks, and examine the extent to which the findings may generalize.
>
> Given these concerns, I will maintain my current score.

---

> ### Author Response · Authors · 2025-08-09
>
> Thank you for the followup comment. We make the following remarks.
> >  the paper examines only the dynamics of a single gradient step, without addressing the full training trajectory
>
> * In our shallow transformer setup, the optimization landscape is non-convex and highly complex, and hence the single-gradient-step simplification is common in the literature – see for example [1,2]. As mentioned in our previous response, insights in these one-step analyses can extend to different gradient-based training algorithms. To our knowledge, existing theoretical works on similar n-gram settings that go beyond the first gradient step all worked with a particular layer-wise training paradigm [3,4,5], and often assumed a known value matrix. Moreover, the novelty of our work among these prior studies lies in characterizing the transformer's algorithm selection from a data-centric perspective — this turns out to be technically nontrivial even in the one GD step setting. We therefore believe that our theoretical analyses should not simply be disregarded as incomplete.
> * To connect our optimization analysis to a more “global” characterization, note that Proposition 5 and Theorem 7 imply that, under a small max-sum ratio, the learned transformer “generalizes OOD”, in the sense of Definition 2: namely, for all OOD input sequences of length up to $L$, the label that receives the highest predicted probability coincides with the correct target token. Hence if the temperature (or equivalently, the scale of $W^V$) is taken to be sufficiently large, the cross-entropy loss becomes arbitrarily close to zero. Consequently, the only remaining gap from (approximate) global optimality lies in the scaling of $W^V$; hence one may interpret our result as suggesting after one GD step, the model has already learned the “global” optimal directions.
> * Last but not least, in Section 4 we have empirically demonstrated a similar transition from failure to success in OOD generalization under standard training with the Adam optimizer. We will include additional experiments that examine the full training dynamics in greater detail (e.g., quantitative assessment of the transition and the learned mechanism).
>
> > the paper should discuss the connection between the studied task and standard NLP tasks
>
> The induction head mechanism has been demonstrated to play a critical role in enabling the in-context learning capabilities of LLMs; many prior theoretical works have studied the gradient-based learning of this mechanism in synthetic copying tasks similar to ours, and as mentioned in our previous response, the understanding of such tasks allows us to potentially infer the model performance on larger and more realistic tasks that involves induction head.
> To our knowledge, our work is the first to characterize how the diversity of data distribution dictates a transition from a spurious positional solution to the “correct” induction head mechanism; while we respect the reviewer's practice-oriented perspective, we believe that our theoretical framing of how the data quality affects the learned algorithm is a nontrivial contribution in itself.
>
> We hope these clarifications are helpful and appreciate your time in engaging with our work.
>
> [1] Bietti et al.  Birth of a Transformer: A Memory Viewpoint, 2023.
>
> [2] Wang and Sato.  Rethinking Associative Memory Mechanism in Induction Head, 2025.
>
> [3] Chen et al.  Unveiling Induction Heads: Provable Training Dynamics and Feature Learning in Transformers, 2024.
>
> [4] Wang et al.  How Transformers Get Rich: Approximation and Dynamics Analysis, 2025.
>
> [5] Nichani et al.  How Transformers Learn Causal Structure with Gradient Descent, 2024.

---

### Official Review · Reviewer_aZH3 · 2025-07-09

**Clarity:** 3
**Significance:** 3
**Originality:** 4
**Rating:** 4
**Confidence:** 4

**Summary:**

This paper investigates how the diversity of training data affects the algorithm a transformer learns in a synthetic in-context learning task. The task involves sequences with two trigger tokens; given a sequence that ends at the second trigger, the model must look back and predict the token that followed the first trigger. Here, diversity in the training set can be considered as variations in the distance and positions of the two triggers across sequences. The test set then evaluates the model’s ability to generalize to longer sequences, i.e., out-of-distribution (OOD) generalization to contexts where the triggers appear later and/or farther apart.

In a simplified theoretical setting with a one-layer transformer and fixed input embeddings, the authors show that a single step of gradient descent can lead the model to adopt one of two strategies: either the so-called induction head mechanism, which copies the token following the first trigger, or a position-based strategy that memorizes fixed post-trigger positions from the training set. Which strategy is learned depends on a *max-sum* ratio defined over the training distribution—a quantity that captures how spread out the trigger positions are.

The authors also present empirical results using a two-layer transformer trained with a standard setup to support their theory: when the training distribution over the trigger positions is narrow, the model fails to generalize. However, increasing the support size of this distribution leads to a sharp transition in OOD generalization

**Questions:**

1. Can you clarify the intuition behind the max-sum ratio? It is true that it captures how pointed or spread the distribution is, but it also depends on the absolute values of the positions in the support. For example, shifting the entire distribution while preserving its shape could change the max-sum value and potentially affect generalization (based on the theory), despite the diversity remaining the same.

2. Regarding experiments in Fig 1:

    a) In some setting with large $l_0$, the model still generalizes even with a narrow distribution (e.g., $l_0=7, k=1$). So, also related to the previous question, the model might generalize even if the data is not very diverse?

    b) Since the theory is based on a single gradient step, does the length of training affect the results and the phase transition?

   c) The theory suggests two mechanisms for prediction: induction head vs. position-based memorization. In the figure, you're just measuring loss to evaluate generalization. Is it possible to also check the mechanism? For example, for induction head, measuring the prediction accuracy (instead of loss) verifies if the correct copying mechanism is implemented. Is there a way to directly measure how much the model relies on position information—especially in the left side of Figure 1, where the model performs poorly?

3. Minor typos: 1) $M_{QK}, M_V$ are not defined in text, 2) (line 126) $W_{KQ}\in\mathbb{R}^{D\times D}$, 3) In Figure 1, using both $i$ and $l_0$ is confusing.

**Ethical Concerns:**

["NO or VERY MINOR ethics concerns only"]

**Final Justification:**

My main concern, the lack of empirical verification for the proposed mechanism, was resolved by the new experiments discussed in the rebuttal. Separately, the expanded discussion of the max-sum ratio was helpful. The study’s design and theoretical message are solid, so I lean toward acceptance. I keep the rating at borderline, however, because elements such as the max-sum ratio diversity metric seem specific to the toy setup and may not generalize to other setups, even though the broader message appears extensible.

**Quality:**

3

**Strengths And Weaknesses:**

The paper introduces a neat and minimal synthetic setup for theoretically studying the impact of training data distribution. The main takeaways—that data diversity helps the model choose the right prediction mechanism, and that there is a trade-off between generalization and statistical complexity depending on the training distribution—are intuitive and nicely supported by the theory. The proof sketches are also helpful in breaking down the main argument.

The theoretical transformer model is simplified, but most of the simplifications—such as merging key-query weights and removing the FFN—are standard in similar theoretical works and are reasonable given the paper’s focus.

A key limitation is that the diversity measure used in the analysis, the max-sum ratio, seems very specific to this task and not easily generalizable. It also depends on absolute position values (see Question 1), which makes its interpretation less straightforward. Please, also see the questions.

---

> ### Author Rebuttal · Authors · 2025-07-31
>
> We thank the reviewer for the helpful feedback. We will correct the typos in the manuscript.  We address the comments and questions below.
>
> ### Intuition behind the max-sum ratio
> As you pointed out, the max-sum ratio is not shift-invariant, since the inverse of the context length — measured by $\ell^{-1}$ — varies under a global shift of the distribution.  For instance, if we consider a uniform distribution over an interval of width $K$, namely $[\ell_0, \ell_0 + 1, \ldots, \ell_0 + K - 1]$, and we shift this distribution to the right (i.e., toward longer contexts) while keeping the total width $K$ fixed, the max-sum ratio decreases. As suggested by Theorem 7, a smaller max-sum ratio is favorable for OOD generalization. Therefore, shifting the distribution to the right reduces the required width $K$ for achieving OOD generalization.
>   The intuition is that the task distribution, when reweighted by the inverse of the context length (i.e., $q_\ell$ weighted by $\ell^{-1}$), should be approximately uniform in order for the max-sum ratio to attain its minimum.  In other words, $q_\ell$ associated with smaller $\ell$—which corresponds to a task where positional information can be memorized easily—should take relatively smaller values.
>
>   A more precise interpretation of the max-sum ratio, then, is that it primarily encourages diversity in the task distribution, while assigning greater weight to longer contexts where positional signals are harder to obtain. In light of your feedback, we plan to incorporate this precise implication into the paper.
>
> ### Regarding experiments in Fig 1
> We address each of your points in the following:
>
> a) As mentioned above, the wider the rightward shift of the distribution over the period length $\ell$ during pretraining, the narrower the required support of the pretraining distribution becomes for implementing an induction head. Thus, this point is consistent with the prediction made by the max-sum ratio.
>
> b) Our learning setup — based on one gradient descent step with a large learning rate — is intended to model the behavior of parameters escaping from saddle points during the initial phase of gradient-based algorithms. When using other algorithms such as one-pass SGD, we expect that the length of pretraining influences the implemented algorithm. This is analogous to the compatibility observed in some feature learning settings between the sample complexity of one-step GD and the computational complexity of one-pass SGD [1,2].
>
> [1] Abbe, Boix-Adsera, and Misiakiewicz: SGD learning on neural networks: leap complexity and saddle-to-saddle dynamics.
>
> [2] Barak et al: Hidden Progress in Deep Learning: SGD Learns Parities Near the Computational Limit.
>
> c) One method to internally verify the mechanism by which each algorithm is implemented in the transformer is to examine the heatmap of the key-query matrix that governs the attention scores. Theoretically, this is formalized in Equation (3.1) (more precisely, Lemmas 10 and 11 in Appendix B), which show that the key-query matrix exhibits peaks corresponding to each algorithm (positional memorization and induction head). Furthermore, using the same data model and algorithm as in our theory, we conducted experiments demonstrating that both peaks indeed appear in the key-query matrix, and that increasing the width of the pretraining task leads to the selective attenuation of the positional memory peak.  We will include these experimental results in the revision.
>
> We would be happy to answer any follow-up questions in the discussion period.

---

> > ### Comment · Reviewer_aZH3 · 2025-08-06
> >
> > Thank you for the additional clarifications.
> >
> > My questions are largely addressed. The new experiments in part (c) are a useful addition because they tighten the connection between the theoretical message and the empirical evidence. I'm inclined to accept the paper, and I will maintain my original score.

---

### Note · Authors · 2025-08-14

We deeply appreciate all the reviewers and the Area Chair for thoroughly engaging in the evaluation and for providing constructive feedback. We are especially grateful for the recognition of our original contribution in providing a clear mechanistic story and solid theoretical analysis of *how positional signals "diffuse" with diverse data while content-based signals "concentrate", leading to the emergence of one mechanism over the other* (Reviewers aZH3, DDFp, eKuv).

During the rebuttal, we:
- Articulated the significance of our theoretical setting on the copying task as a building block for a data-centric approach to larger and more realistic tasks requiring OOD generalization, and motivated the simplified theoretical settings, including single gradient update, merged key–query matrix, and previous-token embedding.
- Clarified our experimental setting, and highlighted that the memorization–generalization transition is also observed in natural transformer training with Adam optimizer  (which indicates that the reported phenomenon is not merely an artifact of our theoretical setting).
- Explained the role of the max-sum ratio, noting in particular that smaller ratios are achieved not only by broadening the pretraining distribution but also by emphasizing longer contexts.

To improve the final version, we will:
- Extend our analysis to multiple triggers, where the dispersion of algorithmic signals require higher data diversity for the transition and increased sample complexity.
- Revise the experimental descriptions for clarity and readability.
- Conduct additional experiments to
  - visualize the learned key-query matrix to illustrate the transition between mechanisms (induction heads and positional memory);
  - $(i)$ further examine the sample complexity under standard training, and $(ii)$ quantify the sharp transition point between the learned mechanisms.
- Expand the related works section to discuss additional references mentioned by the reviewers.

We believe the final version of our paper will benefit greatly from these suggestions and will serve as a meaningful contribution to the community’s understanding of how the data distribution affects the algorithm selection in transformer training.

---

### Decision · Program_Chairs · 2025-09-17

**Decision:**

Accept (spotlight)

**Comment:**

*Main Contribution*
This paper investigates the influence of pre-training data on the test-time computation of language models. Its primary contribution is to establish a phase transition between data memorization and a form of test-time generalization (commonly referred to as induction heads). This transition is governed by the “max–sum” ratio, which quantifies input diversity. Importantly, the analysis is restricted to training with a single iteration of gradient descent.

*Strength*
The work introduces a highly interesting theoretical question and proposes a quantitative metric for characterizing a specific phase transition in test-time computation.

*Weakness*
The main limitation lies in analyzing only a single gradient descent step, rather than studying convergence. This restriction stems from the complexity of the problem. Nonetheless, analyzing a single step can be seen as an important first step toward a deeper theoretical understanding of LLMs.

*Rebuttal*
Most reviewers were convinced by the authors’ response. However, one reviewer maintained that restricting the analysis to a single gradient descent step is a significant limitation.

*Decision*
In my view, the theoretical results presented in this paper are both insightful and novel, and they open a promising new direction for theoretical research on understanding LLMs.

*AC Comment*
There are prior studies examining the interplay between training data distribution and in-context algorithm selection. In particular, several recent works show that ill-conditioned training data can encourage the learning of preconditioned gradient descent for in-context linear regression. The paper may benefit from reviewing these related works.